# Conditional Sampling of Variational Autoencoders via Iterated Approximate Ancestral Sampling

**Vaidotas Simkus** *vaidotas.simkus@ed.ac.uk*
**Michael U. Gutmann** *michael.gutmann@ed.ac.uk*
*School of Informatics*
*University of Edinburgh*

Reviewed on OpenReview: *https://openreview.net/forum?id=I5sJ6PU6JN*

## Abstract

Conditional sampling of variational autoencoders (VAEs) is needed in various applications, such as missing data imputation, but is computationally intractable. A principled choice for asymptotically exact conditional sampling is Metropolis-within-Gibbs (MWG). However, we observe that the tendency of VAEs to learn a structured latent space, a commonly desired property, can cause the MWG sampler to get "stuck" far from the target distribution. This paper mitigates the limitations of MWG: we systematically outline the pitfalls in the context of VAEs, propose two original methods that address these pitfalls, and demonstrate an improved performance of the proposed methods on a set of sampling tasks.

## 1 Introduction

Conditional sampling of modern deep probabilistic models is an important but generally intractable problem. Variational autoencoders (VAEs, Kingma & Welling, 2013; Rezende et al., 2014) are a family of deep probabilistic models that *capture the complexity of real-world data distributions via a structured latent space*. The impressive modelling capability and the usefulness of the structured latent space make VAEs a model of choice in a broad range of domains from healthcare (Han et al., 2019) and chemistry (Gómez-Bombarelli et al., 2018) to images (Child, 2021) and audio (van den Oord et al., 2017). Ancestral sampling can be used for efficient unconditional sampling of VAEs, but many downstream tasks, for example, prediction or missing data imputation (e.g. Goodfellow et al., 2016, Chapter 5.1.1), instead require *conditional sampling*. However, *for VAEs, this is intractable*, and hence approximate methods are needed.

A canonical approximate method is Markov chain Monte Carlo (MCMC, e.g. Barber, 2017, Chapter 27.4) but the general lack of knowledge about the learnt VAE may make tuning, for example, picking a good proposal distribution, and hence successfully using MCMC samplers challenging. To make sampling easier, an approach called Metropolis-within-Gibbs (MWG, Mattei & Frellsen, 2018) re-uses the encoder, an auxiliary component from the training of the VAE, to construct a suitable proposal distribution in a Metropolis–Hastings-type algorithm (Metropolis et al., 1953; Hastings, 1970). The simplicity of MWG and its asymptotic convergence guarantees make it a compelling choice for conditional sampling of VAEs.

While a structured latent space is often a desirable property of VAEs, enabling the modelling of complex distributions, we *notice that this latent structure can cause the Markov chains of MWG to get "stuck"* hence impeding conditional sampling. In this paper we

- Detail the potential pitfalls of Metropolis-within-Gibbs in the context of VAEs (section 3).

- Propose a modification of MWG, called adaptive collapsed-Metropolis-within-Gibbs (AC-MWG, section 4.1), that mitigates the outlined pitfalls and prove its convergence.

- Introduce an alternative sampling method, called latent-adaptive importance resampling (LAIR, section 4.2), which demonstrates an improved sampling performance in our experiments.

- Evaluate the samplers on a set of conditional sampling tasks: (semi-)synthetic, where sampling from the ground truth conditional distributions is computationally tractable, and real-world missing data imputation tasks, where the ground truth distribution is not available.

With the proposed methods we address the conditional sampling problem of VAEs, a key challenge to downstream application of this flexible family of models. Our methods build and improve upon the limitations of MWG enabling more accurate use of VAEs in important tasks like missing data imputation.

## 2 Background: Conditional sampling of VAEs

We here describe the conditional sampling problem and the existing Gibbs-like methods that have been used to draw conditional samples.

### 2.1 Problem and assumptions

Given a pre-trained variational autoencoder, whose generative model we denote as $p(\boldsymbol{x}, \boldsymbol{z}) = p(\boldsymbol{x} \mid \boldsymbol{z})p(\boldsymbol{z})$, where $\boldsymbol{x} = (\boldsymbol{x}_{\text{obs}}, \boldsymbol{x}_{\text{mis}})$ are the visible and $\boldsymbol{z}$ are the latent variables, we would like to sample:

$$p(\boldsymbol{x}_{\text{mis}} \mid \boldsymbol{x}_{\text{obs}}) = \frac{\int p(\boldsymbol{x}_{\text{obs}}, \boldsymbol{x}_{\text{mis}}, \boldsymbol{z}) \, \mathrm{d}\boldsymbol{z}}{p(\boldsymbol{x}_{\text{obs}})} = \int p(\boldsymbol{x}_{\text{mis}} \mid \boldsymbol{x}_{\text{obs}}, \boldsymbol{z})p(\boldsymbol{z} \mid \boldsymbol{x}_{\text{obs}}) \, \mathrm{d}\boldsymbol{z}. \tag{1}$$

The variables $\boldsymbol{x}_{\text{mis}}$ and $\boldsymbol{x}_{\text{obs}}$ are respectively the target/missing and conditioning/observed variables. This choice of notation is motivated by the correspondence between conditional sampling and probabilistic imputation of missing data (Rubin, 1987; 1996).[1] Unlike unconditional generation, *ancestral sampling of $p(\boldsymbol{x}_{mis} \mid \boldsymbol{x}_{obs})$ is generally intractable since the posterior distribution $p(\boldsymbol{z} \mid \boldsymbol{x}_{obs})$ is not accessible* and hence approximations are required.

In the rest of the paper we assume that the generative model is such that computation of $p(\boldsymbol{x}_{\text{obs}} \mid \boldsymbol{z})$ and sampling of $p(\boldsymbol{x}_{\text{mis}} \mid \boldsymbol{x}_{\text{obs}}, \boldsymbol{z})$ is tractable. This is typically the case for most VAE architectures due to conditional independence assumptions (i.e. $x_j \perp\!\!\!\perp \boldsymbol{x}_{\searrow j} \mid \boldsymbol{z}$ for all $\forall j$) or the use of a Gaussian family for the decoder distribution $p(\boldsymbol{x} \mid \boldsymbol{z})$. Moreover, we assume that the encoder distribution, or the amortised variational posterior, $q(\boldsymbol{z} \mid \boldsymbol{x})$ (Gershman & Goodman, 2014) which approximates the model posterior $p(\boldsymbol{z} \mid \boldsymbol{x})$, is available.[2]

### 2.2 Pseudo-Gibbs (Rezende et al., 2014)

Rezende et al. (2014, Appendix F) have proposed a procedure related to Gibbs sampling (Geman & Geman, 1984), also called pseudo-Gibbs (Heckerman et al., 2000; Mattei & Frellsen, 2018), that due to its generality and simplicity has been regularly used for missing data imputation with VAEs (e.g. Rezende et al., 2014; Li et al., 2016; 2017; Rezende et al., 2018; Boquet et al., 2019). Starting with some random imputations $\boldsymbol{x}_{\text{mis}}^0$ the procedure iteratively samples latents $\boldsymbol{z}^t \sim q(\boldsymbol{z} \mid \boldsymbol{x}_{\text{obs}}, \boldsymbol{x}_{\text{mis}}^{t-1})$ and imputations $\boldsymbol{x}_{\text{mis}}^t \sim p(\boldsymbol{x}_{\text{mis}} \mid \boldsymbol{x}_{\text{obs}}, \boldsymbol{z}^t)$.[3] This iterative procedure generates a Markov chain that subject to some conditions on the closeness of the variational posterior $q(\boldsymbol{z} \mid \boldsymbol{x}_{\text{obs}}, \boldsymbol{x}_{\text{mis}})$ and the intractable model posterior $p(\boldsymbol{z} \mid \boldsymbol{x}_{\text{obs}}, \boldsymbol{x}_{\text{mis}})$ converges asymptotically in $t$ to a distribution that approximately follows $p(\boldsymbol{x}_{\text{mis}}, \boldsymbol{z} \mid \boldsymbol{x}_{\text{obs}})$ (Rezende et al., 2014, Proposition F.1). The sampler corresponds to an exact Gibbs sampler if $q(\boldsymbol{z} \mid \boldsymbol{x}_{\text{obs}}, \boldsymbol{x}_{\text{mis}}) = p(\boldsymbol{z} \mid \boldsymbol{x}_{\text{obs}}, \boldsymbol{x}_{\text{mis}})$.

However, the equality $q(\boldsymbol{z} \mid \boldsymbol{x}_{\text{obs}}, \boldsymbol{x}_{\text{mis}}) = p(\boldsymbol{z} \mid \boldsymbol{x}_{\text{obs}}, \boldsymbol{x}_{\text{mis}})$ generally does not hold due to at least one of the following issues: insufficient flexibility of the variational distributional family, amortisation gap, or inference generalisation gap (Cremer et al., 2018; Zhang et al., 2021). Hence, pseudo-Gibbs sampling may produce sub-optimal samples even in the asymptotic limit or completely fail to converge due to an incompatibility of $q(\boldsymbol{z} \mid \boldsymbol{x}_{\text{obs}}, \boldsymbol{x}_{\text{mis}})$ and $p(\boldsymbol{x}_{\text{mis}} \mid \boldsymbol{x}_{\text{obs}}, \boldsymbol{z})$.

---

[1]Equation (1) corresponds directly to missing data imputation with missing-at-random (MAR) missingness pattern.

[2]The variational posterior is typically available after fitting the VAE on complete data using standard variational Bayes (Rezende et al., 2014; Kingma et al., 2014), or can be fitted afterwards using a real or generated complete data set.

[3]Superscript $t$ represents the sampler iteration.

### 2.3 Metropolis-within-Gibbs (Mattei & Frellsen, 2018)

Mattei & Frellsen (2018, Section 3.2) have proposed a simple modification of the pseudo-Gibbs sampler that can asymptotically in $t$ generate exact samples from $p(\boldsymbol{x}_{\mathrm{mis}} \mid \boldsymbol{x}_{\mathrm{obs}})$. The method incorporates a Metropolis–Hastings accept-reject step (Metropolis et al., 1953; Hastings, 1970) to correct for the mismatch between $q(\boldsymbol{z} \mid \boldsymbol{x}_{\mathrm{obs}}, \boldsymbol{x}_{\mathrm{mis}})$ and $p(\boldsymbol{z} \mid \boldsymbol{x}_{\mathrm{obs}}, \boldsymbol{x}_{\mathrm{mis}})$ followed by sampling from $p(\boldsymbol{x}_{\mathrm{mis}} \mid \boldsymbol{x}_{\mathrm{obs}}, \boldsymbol{z})$, hence yielding a sampler in the Metropolis-within-Gibbs (MWG) family (Gelman & Rubin, 1992, Section 4.4). Specifically, at each iteration $t$ it generates the proposal sample $\tilde{\boldsymbol{z}} \sim q(\boldsymbol{z} \mid \boldsymbol{x}_{\mathrm{obs}}, \boldsymbol{x}_{\mathrm{mis}}^{t-1})$ and accepts it as $\boldsymbol{z}^t = \tilde{\boldsymbol{z}}$ with probability

$$\rho^t(\tilde{\boldsymbol{z}}, \boldsymbol{z}^{t-1}; \boldsymbol{x}_{\mathrm{mis}}^{t-1}) = \min \left\{ 1, \frac{p(\boldsymbol{x}_{\mathrm{obs}}, \boldsymbol{x}_{\mathrm{mis}}^{t-1} \mid \tilde{\boldsymbol{z}}) p(\tilde{\boldsymbol{z}})}{p(\boldsymbol{x}_{\mathrm{obs}}, \boldsymbol{x}_{\mathrm{mis}}^{t-1} \mid \boldsymbol{z}^{t-1}) p(\boldsymbol{z}^{t-1})} \frac{q(\boldsymbol{z}^{t-1} \mid \boldsymbol{x}_{\mathrm{obs}}, \boldsymbol{x}_{\mathrm{mis}}^{t-1})}{q(\tilde{\boldsymbol{z}} \mid \boldsymbol{x}_{\mathrm{obs}}, \boldsymbol{x}_{\mathrm{mis}}^{t-1})} \right\}. \tag{2}$$

If the proposal $\tilde{\boldsymbol{z}}$ is rejected, the latent sample from the previous iteration is used, so that $\boldsymbol{z}^t = \boldsymbol{z}^{t-1}$. Given $\boldsymbol{z}^t$, a new imputation $\boldsymbol{x}_{\mathrm{mis}}^t$ is then sampled as in standard Gibbs sampling: $\boldsymbol{x}_{\mathrm{mis}}^t \sim p(\boldsymbol{x}_{\mathrm{mis}} \mid \boldsymbol{x}_{\mathrm{obs}}, \boldsymbol{z}^t)$. By incorporating the Metropolis–Hastings acceptance step, the pseudo-Gibbs sampler is transformed into an asymptotically exact MCMC sampler with $p(\boldsymbol{x}_{\mathrm{mis}}, \boldsymbol{z} \mid \boldsymbol{x}_{\mathrm{obs}})$ as stationary distribution even if $q(\boldsymbol{z} \mid \boldsymbol{x}_{\mathrm{obs}}, \boldsymbol{x}_{\mathrm{mis}}) \neq p(\boldsymbol{z} \mid \boldsymbol{x}_{\mathrm{obs}}, \boldsymbol{x}_{\mathrm{mis}})$.

Importantly, as noted by the authors, the asymptotic exactness of MWG comes, compared to the pseudo-Gibbs sampler, at little additional computational cost in each iteration: the quantities required for computing $\rho^t$ are also computed in the pseudo-Gibbs sampler, except for the often cheap prior evaluations $p(\boldsymbol{z})$.

In summary, MWG has several desirable properties which make it an attractive choice for conditional sampling of VAEs: (i) it provides theoretical guarantees of convergence to the correct conditional distribution, (ii) it is simple to implement, and (iii) its per-iteration computational cost is relatively small, i.e. one standard evaluation of a VAE, and is comparable to the cost of pseudo-Gibbs. However, as we will see next, MWG is not free of important pitfalls.

## 3 Pitfalls of Gibbs-like samplers for VAEs

Although the Gibbs-like samplers from sections 2.2 and 2.3 are often used to conditionally sample from a VAE model, the structure of the latent space can cause poor non-asymptotic sampling behaviour. We here detail, in a form of three pitfalls, how this structure can affect the aforementioned samplers. While the reported pitfalls are related to the known limitations of the classical Gibbs (Geman & Geman, 1984) and Metropolis-within-Gibbs samplers (Gelman & Rubin, 1992), we here work out their significance in the context of VAEs. In fig. 1 we exemplify these pitfalls in an archetypical scenario using a synthetic 2-dimensional VAE model (for details about the model see appendix C.1).[4] The proposed methods in the following section, AC-MWG (section 4.1) and LAIR (section 4.2), provide remedies for the reported pitfalls.

**Pitfall I. Strong relationship between the latents and the visibles can cause poor mixing.** We often train VAEs to learn a structured latent space that captures the complexity of the data. This is typically achieved by using a decoder with a simple, often conditionally-independent, distribution. For example, to fit a binarised MNIST data set well with a Bernoulli decoder distribution $p(\boldsymbol{x} \mid \boldsymbol{z}) = \prod_d \mathrm{Bernoulli}(x_d \mid \boldsymbol{z})$, the digits in the image space must be well-represented in the latent space and the variance of the decoder must be nearly 0, otherwise the model would produce noisy samples due to random "flips" of the pixels. Hence, in VAEs with simple decoders the complexity of modelling the visibles $\boldsymbol{x}$ is often converted to learning a complex structure in the latent space along with a near-deterministic mapping between the latents $\boldsymbol{z}$ and the visibles $\boldsymbol{x}$ as given by the decoder $p(\boldsymbol{x} \mid \boldsymbol{z})$. But this strong, near-deterministic, relationship can substantially inhibit the convergence and mixing properties of a sampler like Metropolis-within-Gibbs. This is because the proposed samples $\tilde{\boldsymbol{z}} \sim q(\boldsymbol{z})$ will be rejected with a high probability if the conditional distribution $p(\boldsymbol{x}_{\mathrm{mis}} \mid \boldsymbol{x}_{\mathrm{obs}}, \tilde{\boldsymbol{z}}) \propto p(\boldsymbol{x}_{\mathrm{obs}}, \boldsymbol{x}_{\mathrm{mis}} \mid \tilde{\boldsymbol{z}})$ places little density/mass on the *previous* value of $\boldsymbol{x}_{\mathrm{mis}} = \boldsymbol{x}_{\mathrm{mis}}^{t-1}$, as a small value of $p(\boldsymbol{x}_{\mathrm{obs}}, \boldsymbol{x}_{\mathrm{mis}}^{t-1} \mid \tilde{\boldsymbol{z}})$ will make the Metropolis–Hastings acceptance probability in eq. (2) small. This small acceptance probability leads to Markov chains that get "stuck" in a mode and prevents the sampler

---

[4]We note that the variational distribution $q(\boldsymbol{z} \mid \boldsymbol{x})$ in this section is constructed to be slightly wider than the model conditional $p(\boldsymbol{z} \mid \boldsymbol{x})$ to differentiate the different modes of failure.

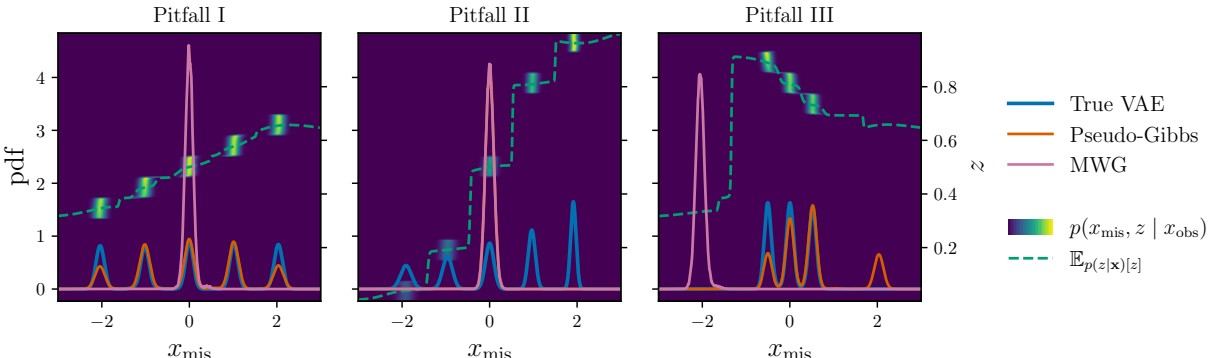

Figure 1: *Pitfalls of Gibbs-like samplers for VAE models.* (The figure is best viewed in colour.) Each panel corresponds to a distinct sampling problem, where the the observed variable $x_{\text{obs}} \in \{x_0, x_1\}$ is, from left to right, $x_1 = 0$, $x_0 = 0$, and $x_1 = 1$. The line plots show the ground truth density $p(x_{\text{mis}} \mid x_{\text{obs}})$ (blue) and the density of the samples obtained from the two Gibbs-like methods, pseudo-Gibbs (orange) and MWG (pink). The contour plot shows the conditional joint density $p(x_{\text{mis}}, z \mid x_{\text{obs}})$ of the VAE model over the missing variable $x_{\text{mis}}$ (bottom axis) and the latent $z$ (right axis), and the dashed green curve shows the expected value of $z$ given $x_0$ and $x_1$. Both samplers were initialised with the same state and run for 50k iterations. *Left:* MWG fails to mix between nearby modes (in the space of $z$; right axis) due to high rejection probability in eq. (2). *Center:* both pseudo-Gibbs and MWG fail to find modes that are far apart (in the space of $z$; right axis) due to narrow proposal distribution. (We note that MWG and pseudo-Gibbs lines overlap in this plot.) *Right:* poor initialisation may leave MWG "stuck" far from the target distribution. Appendix D.1 contains an additional view of the pitfalls.

from moving to nearby modes that are close in the latent space. We illustrate this pitfall in fig. 1 (left). In this example, MWG (pink) fails to mix between the modes that are close in the space of latents. This failure occurs despite the proposal distribution generating samples from the neighbouring modes because such proposed samples are rejected by the Metropolis–Hastings step. On the other hand, pseudo-Gibbs (orange) can mix between the modes since it does not use the Metropolis–Hastings step.

**Pitfall II. The encoder distribution generates proposals that are insufficiently exploratory.** A further complication of the structured latent space is illustrated in fig. 1 (center). Here, the modes of the target distribution are sparsely dispersed in the latent space. In this example, we see that both MWG (pink) and pseudo-Gibbs (orange) fail to find distant modes. This is because the proposal distribution, as given by the encoder that approximates the model posterior $p(\boldsymbol{z} \mid \boldsymbol{x})$, is too "narrow" to propose values from the alternative modes. For example, given the upper-half of an MNIST image of number "8", it may not be possible to tell if the completed image should be an "8" or a "9", representing two modes of imputations. If the latent space representation of "8" and "9" are sufficiently far, then an encoder conditioned on a current imputation state, for example, $\boldsymbol{x}_{\text{obs}} \cup \boldsymbol{x}_{\text{mis}}^{t-1} \equiv$ "9", is unlikely to propose a $\tilde{\boldsymbol{z}}$ that would decode into $\tilde{\boldsymbol{x}}_{\text{mis}}$ in the alternative mode, that is, $\boldsymbol{x}_{\text{obs}} \cup \tilde{\boldsymbol{x}}_{\text{mis}} \equiv$ "8". On the other hand, even if the proposal distribution were wide enough to propose jumps to distant modes, MWG would still reject such proposals with high probability due to pitfall I and thus prevent effective exploration.

**Pitfall III. Poor initialisation can cause sampling of the wrong mode.** As noted by Mattei & Frellsen (2018) MWG for VAEs is extremely sensitive to initialisation, and to alleviate this they suggest initialising by first sampling using pseudo-Gibbs before switching to MWG. But, deciding when to stop the "warm-up" is not easy, and poor initialisation can make MWG get stuck. Moreover, initialisation via an (approximate) MAP using stochastic gradient ascent may also suffer from the multimodality issues described above. In fig. 1 (right) we demonstrate a case where MWG (pink) fails due to a poor initialisation.

The limitations of Gibbs-like samplers described in pitfalls I-III motivate our development of improved samplers. Interestingly, despite pseudo-Gibbs being theoretically inferior to MWG, we have seen in this section that pseudo-Gibbs can under some conditions perform better than MWG (fig. 1). In the following sections we propose two different methods that, like pseudo-Gibbs and MWG, utilise the encoder of the VAE to propose transitions in the latent space, whilst mitigating pitfalls I-III and having stronger theoretical guarantees than the simple pseudo-Gibbs method.

## 4 Remedies

The Metropolis-within-Gibbs (MWG) sampler for conditional sampling of VAEs has several desirable properties (see section 2.3). However, as discussed in the previous section, the Gibbs-like sampler can have poor non-asymptotic performance. In this section we propose two methods for conditional sampling of VAEs inspired by MWG that also mitigate its potential pitfalls (section 3). The key idea of the proposed methods is akin to ancestral sampling of eq. (1); first, the methods approximately sample the intractable posterior over the latents $p(\boldsymbol{z} \mid \boldsymbol{x}_{\text{obs}})$, improve this approximation iteratively, and then sample from the decoder distribution $p(\boldsymbol{x}_{\text{mis}} \mid \boldsymbol{x}_{\text{obs}}, \boldsymbol{z})$ conditional on the produced latent samples. In section 4.1 we propose a few simple modifications to the MWG sampler and demonstrate on a synthetic example how this mitigates the pitfalls of MWG. In section 4.2 we propose an alternative method based on adaptive importance sampling and likewise demonstrate on a synthetic example how it mitigates the pitfalls of MWG. Detailed evaluation of the proposed methods is provided in section 5 and the code to reproduce the experiments is available at https://github.com/vsimkus/vae-conditional-sampling.

### 4.1 Adaptive collapsed-Metropolis-within-Gibbs

We propose several modifications to the MWG sampler from section 2.3 to mitigate the pitfalls outlined in section 3. The proposed sampler is summarised in algorithm 1.

First, to improve exploration and reduce the effects of poor initialisation (see pitfalls II and III) we introduce a prior–variational mixture proposal[5]

$$\tilde{q}_\epsilon(\boldsymbol{z} \mid \boldsymbol{x}_{\text{obs}}, \boldsymbol{x}_{\text{mis}}) = (1 - \epsilon)q(\boldsymbol{z} \mid \boldsymbol{x}_{\text{obs}}, \boldsymbol{x}_{\text{mis}}) + \epsilon p(\boldsymbol{z}), \tag{3}$$

where $q(\boldsymbol{z} \mid \boldsymbol{x}_{\text{obs}}, \boldsymbol{x}_{\text{mis}})$ is the variational encoder distribution, $p(\boldsymbol{z})$ is the prior distribution of the VAE, and $\epsilon \in (0, 1)$ is the probability to sample from the prior. Clearly this modification alone would not resolve the pitfalls of MWG, since proposals $\tilde{\boldsymbol{z}}$ sampled from the prior $p(\boldsymbol{z})$ would be rejected with high probability at the Metropolis–Hastings step due to disagreement with the current imputation $\boldsymbol{x}_{\text{mis}}^{t-1}$ in eq. (2).

Hence, we next propose changing the target distribution of the Metropolis–Hastings step from $p(\boldsymbol{z} \mid \boldsymbol{x}_{\text{obs}}, \boldsymbol{x}_{\text{mis}})$ to $p(\boldsymbol{z} \mid \boldsymbol{x}_{\text{obs}})$, such that a good proposal $\tilde{\boldsymbol{z}}$ would not be rejected due to a disagreement with an imputation $\boldsymbol{x}_{\text{mis}}^{t-1}$ (see pitfall I). The modified Metropolis–Hastings acceptance probability is defined as

$$\rho^t(\tilde{\boldsymbol{z}}^t, \boldsymbol{z}^{t-1}; \tilde{\boldsymbol{x}}_{\text{mis}}) = \min\left\{1, \frac{p(\boldsymbol{x}_{\text{obs}} \mid \tilde{\boldsymbol{z}}^t)p(\tilde{\boldsymbol{z}}^t)}{p(\boldsymbol{x}_{\text{obs}} \mid \boldsymbol{z}^{t-1})p(\boldsymbol{z}^{t-1})} \frac{\tilde{q}_\epsilon(\boldsymbol{z}^{t-1} \mid \boldsymbol{x}_{\text{obs}}, \tilde{\boldsymbol{x}}_{\text{mis}})}{\tilde{q}_\epsilon(\tilde{\boldsymbol{z}}^t \mid \boldsymbol{x}_{\text{obs}}, \tilde{\boldsymbol{x}}_{\text{mis}})}\right\}. \tag{4}$$

Marginalising the missing variables $\boldsymbol{x}_{\text{mis}}$ out of the likelihood $p(\boldsymbol{x}_{\text{obs}}, \boldsymbol{x}_{\text{mis}} \mid \tilde{\boldsymbol{z}}^t)$ corresponds to reducing the conditioning (or collapsing) in Gibbs samplers which is a common approach to improve mixing and convergence (van Dyk & Park, 2008; van Dyk & Jiao, 2015). In our case, if the optimal proposal distribution $p(\boldsymbol{z} \mid \boldsymbol{x}_{\text{obs}})$ were known, the sampler would become a standard ancestral sampler and would be maximally efficient, i.e. it would draw an independent sample at each iteration. Moreover, rather than using the imputation $\boldsymbol{x}_{\text{mis}}^{t-1}$ from the previous iteration to condition the proposal distribution, as in MWG, we are here going to re-sample a random imputation $\tilde{\boldsymbol{x}}_{\text{mis}}$ from an available set of historical imputations $\mathcal{H}_{\text{mis}}^{t-1}$ that is updated adaptively with iterations $t$.

We now combine the proposed changes in eqs. (3) and (4) to introduce the algorithm called adaptive collapsed-Metropolis-within-Gibbs (AC-MWG), which can be seen as an instance of the class of adaptive independent

---

[5]Our mixture proposal is related to the small-world proposal of Guan et al. (2006), which has been shown to improve performance in complicated heterogeneous and multimodal distributions.

---

**Algorithm 1** Adaptive collapsed-Metropolis-within-Gibbs

---

**Input:** VAE model $p(\boldsymbol{x}, \boldsymbol{z})$, variational posterior $q(\boldsymbol{z} \mid \boldsymbol{x}_{\text{mis}}, \boldsymbol{x}_{\text{obs}})$, mixture prob. $\epsilon$, and data-point $\boldsymbol{x}_{\text{obs}}$

1: $\mathcal{H}_{\text{mis}}^0 = \varnothing$                $\triangleright$ Initialise imputation history
2: $(\boldsymbol{z}^0, \boldsymbol{x}_{\text{mis}}^0) \sim p(\boldsymbol{z})p(\boldsymbol{x}_{\text{mis}} \mid \boldsymbol{x}_{\text{obs}}, \boldsymbol{z})$          $\triangleright$ Sample the initial values
3: **for** $t = 1$ **to** $T$ **do**
4:      $\tilde{\boldsymbol{x}}_{\text{mis}} \sim \text{Uniform}(\mathcal{H}_{\text{mis}}^{t-1})$        $\triangleright$ Choose random $\boldsymbol{x}_{\text{mis}}$ from the history
5:      $\tilde{\boldsymbol{z}} \sim \tilde{q}_\epsilon(\boldsymbol{z} \mid \boldsymbol{x}_{\text{obs}}, \tilde{\boldsymbol{x}}_{\text{mis}})$          $\triangleright$ Sample proposal value $\tilde{\boldsymbol{z}}$
6:      $\rho^t = \rho^t(\tilde{\boldsymbol{z}}, \boldsymbol{z}^{t-1}; \tilde{\boldsymbol{x}}_{\text{mis}})$       $\triangleright$ Calculate acceptance probability using eq. (4)
7:      **if** $u < \rho^t$, with $u \sim \text{Uniform}(0, 1)$ **then**       $\triangleright$ Accept $\tilde{\boldsymbol{z}}$ with probability $\rho^t$
8:          $\boldsymbol{z}^t = \tilde{\boldsymbol{z}}$
9:          $\mathcal{H}_{\text{mis}}^t = \{\boldsymbol{x}_{\text{mis}}^\tau\}_{\tau=0}^{t-1}$
10:      **else**                 $\triangleright$ Reject $\tilde{\boldsymbol{z}}$ with probability $\rho^t$
11:          $\boldsymbol{z}^t = \boldsymbol{z}^{t-1}$
12:          $\mathcal{H}_{\text{mis}}^t = \mathcal{H}_{\text{mis}}^{t-1}$
13:      **end if**
14:      $\boldsymbol{x}_{\text{mis}}^t \sim p(\boldsymbol{x}_{\text{mis}} \mid \boldsymbol{x}_{\text{obs}}, \boldsymbol{z}^t)$              $\triangleright$ Sample $\boldsymbol{x}_{\text{mis}}$
15: **end for**
**return** $\{(\boldsymbol{x}_{\text{mis}}^0, \boldsymbol{z}^0), \ldots, (\boldsymbol{x}_{\text{mis}}^T, \boldsymbol{z}^T)\}$          $\triangleright$ Return all samples

---

Metropolis–Hastings algorithms (Holden et al., 2009). Assume we start with an initial latent state $\boldsymbol{z}^0$ and an imputation history $\mathcal{H}_{\text{mis}}^0 = \{\hat{\boldsymbol{x}}_{\text{mis}}^0\}$, such that $\boldsymbol{z}^0$ and $\hat{\boldsymbol{x}}_{\text{mis}}^0$ are mutually independent (for example, $\boldsymbol{z}^0$ and $\hat{\boldsymbol{x}}_{\text{mis}}^0$ are generated via independent short runs of pseudo-Gibbs, see section 2.2, or LAIR, see section 4.2). Then a single iteration $t$ of the sampler is as follows:

1. **Proposal sampling.** First, a historical sample $\tilde{\boldsymbol{x}}_{\text{mis}}$ is re-sampled uniformly at random from the available imputation history $\mathcal{H}_{\text{mis}}^{t-1}$.[6] We then use the proposal distribution from eq. (3) to sample a single proposal $\tilde{\boldsymbol{z}}$.

2. **Metropolis–Hastings acceptance.** The proposed sample $\tilde{\boldsymbol{z}}$ is then either accepted as $\boldsymbol{z}^t = \tilde{\boldsymbol{z}}$ with probability $\rho^t(\tilde{\boldsymbol{z}}, \boldsymbol{z}^{t-1}; \tilde{\boldsymbol{x}}_{\text{mis}})$ in eq. (4) or rejected leaving $\boldsymbol{z}^t = \boldsymbol{z}^{t-1}$.

3. **Imputation sampling.** The imputation $\boldsymbol{x}_{\text{mis}}^t$ is updated by sampling the conditional $\boldsymbol{x}_{\text{mis}}^t \sim p(\boldsymbol{x}_{\text{mis}} \mid \boldsymbol{x}_{\text{obs}}, \boldsymbol{z}^t)$.

4. **Adaptation (history update).** The available history $\mathcal{H}_{\text{mis}}^t$ is updated as follows: if a new $\tilde{\boldsymbol{z}}$ has been accepted then all imputations $\{\boldsymbol{x}_{\text{mis}}^\tau\}_{\tau=0}^{t-1}$ up to step $t-1$ are made available at the next iteration, i.e. $\mathcal{H}_{\text{mis}}^t = \{\boldsymbol{x}_{\text{mis}}^\tau\}_{\tau=0}^{t-1}$, otherwise it is left unchanged $\mathcal{H}_{\text{mis}}^t = \mathcal{H}_{\text{mis}}^{t-1}$.

Step 4 of the sampler constructs the available history $\mathcal{H}_{\text{mis}}^{t-1}$ for the next iteration such that it does not contain imputations that depend on the current state $\boldsymbol{z}^{t-1}$, which ensures that the proposed values $\tilde{\boldsymbol{z}}$ are independent of $\boldsymbol{z}^{t-1}$ and thus guarantees that the stationary distribution of the independent Metropolis–Hastings remains correct as the history $\mathcal{H}_{\text{mis}}^{t-1}$ changes (Roberts & Rosenthal, 2007; Holden et al., 2009). However, the dependence on the sample history $\mathcal{H}_{\text{mis}}^{t-1}$ makes AC-MWG non-Markovian, and hence convergence needs to be verified. Adapting proofs by Holden et al. (2009), we prove in appendix A that the Markov chain of AC-MWG correctly converges to the stationary distribution $p(\boldsymbol{z}, \boldsymbol{x}_{\text{mis}} \mid \boldsymbol{x}_{\text{obs}})$ with probability arbitrarily close to 1 as the number of iterations $T$ grows.

Finally, we note that the per-iteration computational cost of AC-MWG and MWG (section 2.3) are nearly the same. The differences are: re-sampling $\tilde{\boldsymbol{x}}_{\text{mis}}$ from the history $\mathcal{H}_{\text{mis}}^{t-1}$, which should be negligible compared to the cost of evaluating the model, and marginalising the missing variables from the likelihood $p(\boldsymbol{x}_{\text{obs}} \mid \boldsymbol{z}) = \int p(\boldsymbol{x}_{\text{obs}}, \boldsymbol{x}_{\text{mis}} \mid \boldsymbol{z}) \, d\boldsymbol{x}_{\text{mis}}$, which is often free if the standard conditional independence assumption holds.

---

[6]In this paper, we re-sample $\tilde{\boldsymbol{x}}_{\text{mis}}$ from all the past samples in the available history $\mathcal{H}_{\text{mis}}^{t-1}$, however other strategies might be devised to improve the computational and convergence properties of the algorithm (see e.g. Holden et al., 2009; Martino et al., 2018). For example, by using a shorter window of past samples instead of the full length of the history.

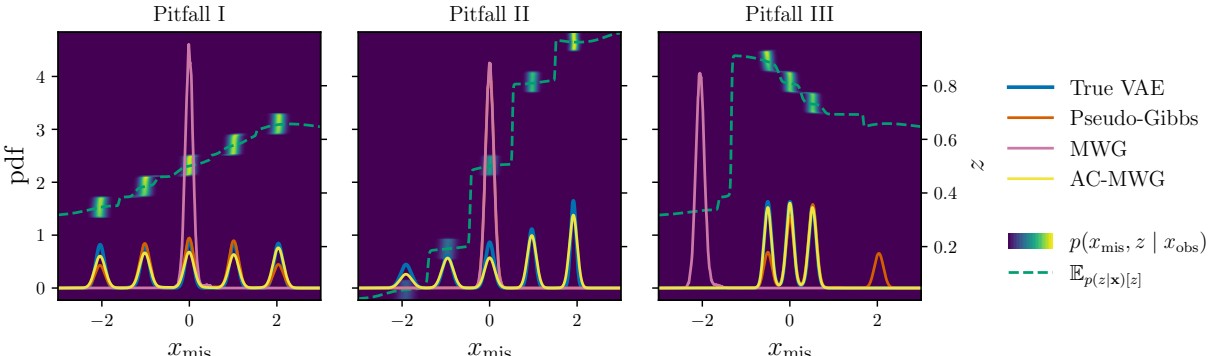

Figure 2: *The proposed AC-MWG sampler (yellow) with $\epsilon = 0.01$ on 2D VAE sampling problems, same as in fig. 1.* (The figure is best viewed in colour.) AC-MWG (yellow) samples the target distribution (blue) more accurately than MWG (pink) and pseudo-Gibbs (orange). All three samplers were initialised with the same state and run for 50k iterations.

### 4.1.1 Verification of AC-MWG on synthetic VAE

We verify the proposed AC-MWG method on the synthetic VAE example in section 3 (see additional details in appendix C.1). The results are shown in fig. 2 (see also additional figures in appendix D.1). With the proposed modifications, AC-MWG samples the target distribution more accurately by exploring modes that are close in the latent space (left) due to the modified acceptance probability in eq. (4), as well as distant modes (center) due to the modified proposal distribution in eq. (3). The modified method is also less sensitive to poor initialisation (right). Moreover, we perform ablation studies in appendices D.2 and D.4 to further validate that both modifications, the mixture proposal in eq. (3) and the collapsed-Gibbs target in eq. (4), are key to the performance of the method.

### 4.2 Latent-adaptive importance resampling

Instead of MCMC, we can sample from eq. (1) via importance resampling (IR, see appendix B for details on standard importance resampling and Chopin & Papaspiliopoulos, 2020, for a comprehensive introduction). However, like MCMC, the efficiency of IR significantly depends on the choice of the proposal distribution. Our goal in this section is to design an *adaptive* importance resampling method that efficiently samples $p(\boldsymbol{x}_{\mathrm{mis}} \mid \boldsymbol{x}_{\mathrm{obs}})$ of a joint VAE model $p(\boldsymbol{x})$, and we achieve this by constructing an adaptive proposal distribution $q^t(\boldsymbol{z} \mid \boldsymbol{x}_{\mathrm{obs}})$ using the encoder distribution $q(\boldsymbol{z} \mid \boldsymbol{x}_{\mathrm{obs}}, \boldsymbol{x}_{\mathrm{mis}})$. The proposed method is summarised in algorithm 2.

As for AC-MWG, we aim to promote exploration and reduce the effects of poor initialisation (see pitfalls II and III). We thus start with the prior–variational mixture proposal $\tilde{q}_\epsilon(\boldsymbol{z} \mid \boldsymbol{x}_{\mathrm{obs}}, \boldsymbol{x}_{\mathrm{mis}})$ from eq. (3) and use it to construct the following *adaptive* mixture proposal distribution $q^t(\boldsymbol{z} \mid \boldsymbol{x}_{\mathrm{obs}})$,

$$q^t(\boldsymbol{z} \mid \boldsymbol{x}_{\mathrm{obs}}) = \mathbb{E}_{f^t(\boldsymbol{x}_{\mathrm{mis}} \mid \boldsymbol{x}_{\mathrm{obs}})} \left[ \tilde{q}_\epsilon(\boldsymbol{z} \mid \boldsymbol{x}_{\mathrm{obs}}, \boldsymbol{x}_{\mathrm{mis}}) \right] \quad \text{with} \quad f^t(\boldsymbol{x}_{\mathrm{mis}} \mid \boldsymbol{x}_{\mathrm{obs}}) = \frac{1}{K} \sum_{k=1}^{K} \delta_{\boldsymbol{x}_{\mathrm{mis}}^{(t-1,k)}}(\boldsymbol{x}_{\mathrm{mis}}),$$

where $f^t(\boldsymbol{x}_{\mathrm{mis}} \mid \boldsymbol{x}_{\mathrm{obs}})$ is an imputation distribution represented as a mixture of Dirac masses at $K$ particles $\{\boldsymbol{x}_{\mathrm{mis}}^{(t-1,k)}\}_{k=1}^{K}$, which we will use to adapt the proposal distribution at each iteration $t$. We further rewrite the proposal by inserting the definition of $f^t(\boldsymbol{x}_{\mathrm{mis}} \mid \boldsymbol{x}_{\mathrm{obs}})$ and $\tilde{q}_\epsilon(\boldsymbol{z} \mid \boldsymbol{x}_{\mathrm{obs}}, \boldsymbol{x}_{\mathrm{mis}})$, and re-parametrise it by setting $\epsilon = \frac{R}{K+R}$, where $R$ is a non-negative integer, to obtain

$$q^t(\boldsymbol{z} \mid \boldsymbol{x}_{\mathrm{obs}}) = \frac{1}{K+R} \left( \sum_{k=1}^{K} q(\boldsymbol{z} \mid \boldsymbol{x}_{\mathrm{obs}}, \boldsymbol{x}_{\mathrm{mis}}^{(t-1,k)}) + \sum_{r=1}^{R} p(\boldsymbol{z}) \right). \tag{5}$$

---

**Algorithm 2** Latent-adaptive importance resampling

---

**Input:** VAE model $p(\boldsymbol{x}, \boldsymbol{z})$, variational posterior $q(\boldsymbol{z} \mid \boldsymbol{x})$, data-point $\boldsymbol{x}_{\text{obs}}$, number of imputation particles $K$, number of iterations $T$

1: $\boldsymbol{x}_{\text{mis}}^{(0,1)}, \ldots, \boldsymbol{x}_{\text{mis}}^{(0,K)} \sim \mathbb{E}_{p(\boldsymbol{z})}\left[p(\boldsymbol{x}_{\text{mis}} \mid \boldsymbol{x}_{\text{obs}}, \boldsymbol{z})\right]$           ▷ Sample the initial imputation particle values

2: **for** $t = 1$ **to** $T$ **do**

3:      $\tilde{\boldsymbol{z}}^{(t,k)} \sim q(\boldsymbol{z} \mid \boldsymbol{x}_{\text{obs}}, \boldsymbol{x}_{\text{mis}}^{(t-1,k)})$ for $\forall k \in \{1, \ldots, K\}$         ▷ Draw a sample for each particle.

4:      $\tilde{\boldsymbol{z}}^{(t,K+r)} \sim p(\boldsymbol{z})$ for $\forall r \in \{1, \ldots, R\}$                   ▷ Draw R prior proposals.

5:      $w(\tilde{\boldsymbol{z}}^{(t,k)}) = \frac{p(\boldsymbol{x}_{\text{obs}}, \tilde{\boldsymbol{z}}^{(t,k)})}{q^t(\tilde{\boldsymbol{z}}^{(t,k)} \mid \boldsymbol{x}_{\text{obs}})}$ for $\forall k \in \{1, \ldots, K+R\}$     ▷ Unnormalised importance weights.

6:      $\tilde{w}(\tilde{\boldsymbol{z}}^{(t,k)}) = \frac{w(\tilde{\boldsymbol{z}}^{(t,k)})}{\sum_{j=1}^{K+R} w(\tilde{\boldsymbol{z}}^{(t,j)})}$ for $\forall k \in \{1, \ldots, K+R\}$       ▷ Normalise importance weights.

7:      $\boldsymbol{z}^{(t,1)}, \ldots, \boldsymbol{z}^{(t,K)} \sim \text{Multinomial}\left(\{\tilde{\boldsymbol{z}}^{(t,k)}, \tilde{w}(\tilde{\boldsymbol{z}}^{(t,k)})\}_{k=1}^{K+R}\right)$    ▷ Resample $\boldsymbol{z}^{(t,k)}$ from the proposed set.

8:      $\boldsymbol{x}_{\text{mis}}^{(t,k)} \sim p(\boldsymbol{x}_{\text{mis}} \mid \boldsymbol{x}_{\text{obs}}, \boldsymbol{z}^{(t,k)})$ for $\forall k \in \{1, \ldots, K\}$.          ▷ Update imputation particles.

9: **end for**

10: $\bar{w}(\tilde{\boldsymbol{z}}^{(t,k)}) = \frac{w(\tilde{\boldsymbol{z}}^{(t,k)})}{\sum_{\tau=1}^{T}\sum_{j=1}^{K+R} w(\tilde{\boldsymbol{z}}^{(\tau,j)})}$ for $\forall k \in \{1, \ldots, K+R\}$ and $\forall t \in \{1, \ldots, T\}$  ▷ Re-norm. all proposals.

11: $\boldsymbol{z}^i \sim \text{Multinomial}\left(\{\tilde{\boldsymbol{z}}^{(t,k)}, \bar{w}(\tilde{\boldsymbol{z}}^{(t,k)})\}_{t=1,k=1}^{(T,K+R)}\right)$ for $\forall i \in \{1, \ldots, T \cdot K\}$  ▷ Resample proposals from all iter.

12: $\boldsymbol{x}_{\text{mis}}^i \sim p(\boldsymbol{x}_{\text{mis}} \mid \boldsymbol{x}_{\text{obs}}, \boldsymbol{z}^i)$ for $\forall i \in \{1, \ldots, T \cdot K\}$.            ▷ Sample imputations

**return** $\{\boldsymbol{x}_{\text{mis}}^i\}_{i=1}^{T \cdot K}$

---

The above proposal can be interpreted to have a total of $K + R$ components of which $K$ components depend on the imputation particles $\{\boldsymbol{x}_{\text{mis}}^{(t-1,k)}\}_{k=1}^{K}$, which encourage exploitation, and $R$ are "replenishing" prior components $p(\boldsymbol{z})$, which encourage exploration and mitigate particle collapse. Moreover, we sample the proposal distribution using stratified sampling (Robert & Casella, 2004; Owen, 2013, Section 9.12; Elvira et al., 2019, Appendix A), a well-known variance-reduction technique that draws one sample from each of the $K + R$ components.

Using the mixture proposal distribution in eq. (5) we now introduce the new algorithm that we call latent-adaptive importance resampling (LAIR), which belongs to the class of adaptive importance sampling algorithms (AIS) of Elvira & Martino (2022, Section 4). The algorithm starts with $K$ imputation particles $\{\boldsymbol{x}_{\text{mis}}^{(0,k)}\}_{k=1}^{K}$ that may come from a simple distribution such as the empirical marginals, another multiple imputation method, or simply the unconditional marginal of the VAE $p(\boldsymbol{x}_{\text{mis}})$. An iteration $t$ of the algorithm then performs the following three steps:

1. **Proposal sampling.** Sample the proposal distribution $q^t(\boldsymbol{z} \mid \boldsymbol{x}_{\text{obs}})$ in eq. (5) using stratified sampling. That is, for each particle $\boldsymbol{x}_{\text{mis}}^{(t-1,k)}$ draw a sample $\tilde{\boldsymbol{z}}^{(t,k)}$ from the proposal $q(\boldsymbol{z} \mid \boldsymbol{x}_{\text{obs}}, \boldsymbol{x}_{\text{mis}}^{(t-1,k)})$ and draw $R$ proposals $\tilde{\boldsymbol{z}}^{(t,K+r)}$ from the prior $p(\boldsymbol{z})$, for a total of $K + R$ proposals.

2. **Weighting.** Compute the unnormalised importance weights $w(\tilde{\boldsymbol{z}}^{(t,k)})$.[7][8]

$$w(\tilde{\boldsymbol{z}}^{(t,k)}) = \frac{p(\boldsymbol{x}_{\text{obs}}, \tilde{\boldsymbol{z}}^{(t,k)})}{q^t(\tilde{\boldsymbol{z}}^{(t,k)} \mid \boldsymbol{x}_{\text{obs}})}. \tag{6}$$

3. **Adaptation.**

   3.I. Resample a set $\{\boldsymbol{z}^{(t,k)}\}_{k=1}^{K}$ with replacement from the proposal set $\{\tilde{\boldsymbol{z}}^{(t,k)}\}_{k=1}^{K+R}$ proportionally to the weights $w(\tilde{\boldsymbol{z}}^{(t,k)})$.[9]

   3.II. Update the imputation particles $\{\boldsymbol{x}_{\text{mis}}^{(t,k)}\}_{k=1}^{K}$ by sampling $p(\boldsymbol{x}_{\text{mis}} \mid \boldsymbol{x}_{\text{obs}}, \boldsymbol{z})$ conditional on each $\boldsymbol{z} \in \{\boldsymbol{z}^{(t,k)}\}_{k=1}^{K}$ from step 3.I.

---

[7]We here use deterministic-mixture MIS (DM-MIS) weights but alternative weighting schemes can also be used that enable a more fine-grained control of the cost-variance trade-off, see Elvira et al. (2019, Section 7.2).

[8]By marginalising the variables $\boldsymbol{x}_{\text{mis}}$ in the numerator of the weights we address pitfall I, similar to eq. (4) of AC-MWG.

[9]Alternative resampling schemes may also be used, see Chopin & Papaspiliopoulos (2020, Section 9.4).

Each iteration $t$ at step 3.II. (accordingly, line 8 of algorithm 2) produces (approximate) samples $\{\boldsymbol{x}_{\text{mis}}^{(t,k)}\}_{k=1}^{K}$ from the target distribution $p(\boldsymbol{x}_{\text{mis}} \mid \boldsymbol{x}_{\text{obs}})$, since any iteration $t$ of the algorithm corresponds to standard importance resampling and hence inherits its properties (Cappé et al., 2004), see appendix B. In particular, the sampler monotonically approaches the target distribution as the number of proposed samples $K + R$ tends to infinity. Hence, the algorithm may be used in settings where the target distribution changes across iterations $t$, for instance, when fitting a model from incomplete data via a Monte Carlo EM (Wei & Tanner, 1990; Simkus et al., 2023).

However, unlike MCMC methods, a finite set of samples at any iteration $t$ are not generally guaranteed to convergence to the target distribution as $t$ grows large (Cappé et al., 2004; Douc et al., 2007). In particular, for finite sample sizes, $K + R \ll \infty$, the sampler bias is of the order $\mathcal{O}(\frac{1}{K+R})$ (Owen, 2013; Paananen et al., 2021) at any iteration $t$ and depends on the disparity between the proposal and the target distributions. To improve the approximation, after the algorithm completes all $T$ iterations, we can use samples from all iterations $t \in \{1, \ldots, T\}$ to construct a more accurate estimator (Cappé et al., 2004).

4. **Draw final samples after completing all $T$ iterations.**

   4.I. Re-normalise the weights of $\tilde{\boldsymbol{z}}^{(t,k)}$ over all iterations $t \in \{0, \ldots, T\}$ and all $k \in \{1, \ldots, K + R\}$ to obtain $\bar{w}(\tilde{\boldsymbol{z}}^{(t,k)}) = \frac{w(\tilde{\boldsymbol{z}}^{(t,k)})}{\sum_{\tau=1}^{T} \sum_{j=1}^{K+R} w(\tilde{\boldsymbol{z}}^{(\tau,j)})}$.

   4.II. Resample $T \cdot K$ samples $\boldsymbol{z}^i$ with replacement from the set $\{\tilde{\boldsymbol{z}}^{(t,k)}\}_{t=1,k=1}^{(T,K+R)}$ using the weights $\bar{w}(\tilde{\boldsymbol{z}}^{(t,k)})$ from the previous step.

   4.III. Sample imputations $\{\boldsymbol{x}_{\text{mis}}^i\}_{i=1}^{T \cdot K}$ via ancestral sampling by sampling $p(\boldsymbol{x}_{\text{mis}} \mid \boldsymbol{x}_{\text{obs}}, \boldsymbol{z})$ conditional on each $\boldsymbol{z} \in \{\boldsymbol{z}^i\}_{i=1}^{T \cdot K}$ from step 4.II.

The advantage of resampling from the re-weighted full sequence of samples is that the bias of the self-normalised importance sampler goes down with $T$ (in addition to $K + R$) and hence more accurate samples can be obtained. In particular, the sampler now monotonically approaches the target distribution as the *total* number of proposed samples approaches infinity, $T(K + R) \to \infty$, and the bias is of the order $\mathcal{O}(\frac{1}{T(K+R)})$.

We note that the per-iteration computational cost of LAIR is comparable to running $K + R$ parallel chains of MWG, with the exception of: marginalising the missing variables from the likelihood, as in AC-MWG, which may often be cheap, and evaluating the denominator of the importance weights $w(\tilde{\boldsymbol{z}})$ in eq. (6), which requires that each of the $K + R$ proposed samples $\tilde{\boldsymbol{z}}$ must be evaluated with the densities of all $K + R$ components in the mixture proposal in eq. (5), hence needing $(K + R)^2$ evaluations. However, since the components of the proposal distribution in eq. (5) are typically all simple distributions, such as a diagonal Gaussians, the computational cost is often negligible for moderate number of proposals $K + R$. Moreover, the cost may be reduced by trading-off for a higher-variance of the estimator, see footnote 7. Finally, the computational cost of the final resampling in step 4 (accordingly, lines 10 to 12 in algorithm 2) is negligible since all the required quantities have already been computed in the past iterations.

### 4.2.1 Verification of LAIR on synthetic VAE

We now verify the proposed method, LAIR, on the synthetic VAE example in section 3 (see additional details in appendix C.1). The results are demonstrated in fig. 3 (see also additional figures in appendix D.1), where we have used $K = 19$ particles and $R = 1$ replenishing components (corresponding to $\epsilon = 0.05$). We can see that the method mitigates the three main pitfalls: poor mixing (left), poor exploration (center), and is less sensible to poor initialisation (right). Moreover, in ablation studies performed in appendices D.2 and D.4 we further investigate the sensitivity of the method to choices of $\epsilon = \frac{R}{K+R}$ and find that the method performs well as long as $0 < \epsilon < 1$.

## 5 Evaluation

In sections 4.1 and 4.2 we have introduced our methods, AC-MWG and LAIR, for conditional sampling of VAEs which mitigate the potential pitfalls of Gibbs-like samplers (section 3) as verified in sections 4.1.1

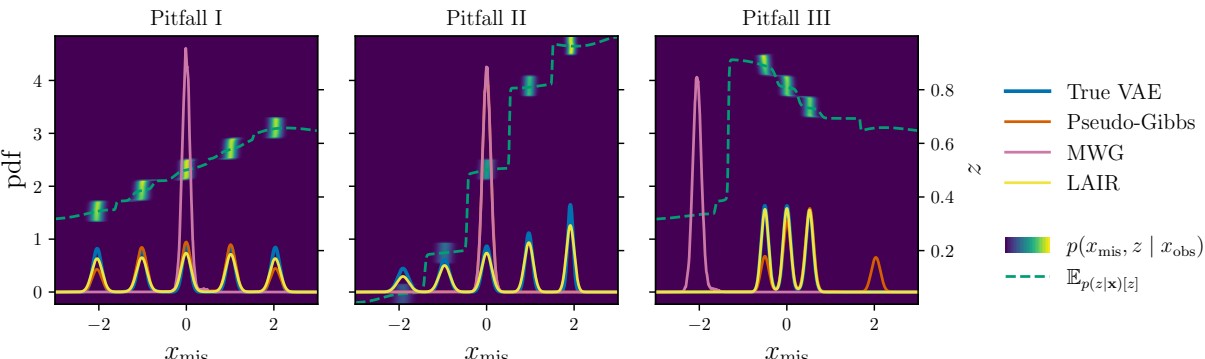

Figure 3: *The proposed LAIR sampler (yellow) with $K = 19$ particles and $R = 1$ replenishing components on 2D VAE sampling problems, same as in fig. 1.* (The figure is best viewed in colour.) LAIR (yellow) samples the target distribution (blue) more accurately than MWG (pink) and pseudo-Gibbs (orange). MWG and pseudo-Gibbs were run for 50k iterations, and LAIR was run for 2.5k iterations to match the number of generative model evaluations.

and 4.2.1. As motivated in section 2.1, conditional sampling is a fundamental tool for multiple imputation of missing data (Rubin, 1987; 1996), where the goal is to generate plausible values of the missing variables with correct uncertainty representation. We here evaluate the newly proposed methods for missing data imputation. We assume that we have a pre-trained VAE model, trained on complete data, and aim to generate imputations of the missing variables at test time.

## 5.1 Mixture-of-Gaussians MNIST

Evaluating the quality of imputations from data alone is a difficult task since the imputations represent guesses of unobserved values from an unknown conditional distribution (Abayomi et al., 2008; van Buuren, 2018, Section 2.5). Hence, to accurately evaluate the proposed methods, in this section we first fit a mixture-of-Gaussians (MoG) model to the MNIST data set, which we then use as the ground truth to simulate a semi-synthetic data set that is subsequently fitted by a VAE model (see appendix C.2 for more details). Using an intermediate MoG model enables us to tractably sample the reference conditional distribution (which would otherwise be unknown) when evaluating the accuracy of the conditional VAE samples obtained using the proposed and existing methods.

In fig. 4 we demonstrate the performance of the methods on 10 sampling problems (see appendix D.3 for additional figures and metrics). We measure the performance using the Fréchet inception distance (FID, Heusel et al., 2017), where for the inception features we use the final layer outputs of the encoder network. The figures show that the proposed methods, AC-MWG (pink) and LAIR (yellow), significantly outperform the performance of the Gibbs-like samplers from sections 2.2 and 2.3 (blue and green). In appendix D.4 we further perform an ablation study for the proposed methods, where: we validate that both the mixture proposal in eq. (3) and the collapsed-Gibbs target in eq. (4) are key to the good performance of AC-MWG; and we find that LAIR can perform well for a number of values of $\epsilon = \frac{R}{K+R}$, as long as $0 < \epsilon < 1$.

The results for MWG (green) use pseudo-Gibbs warm-up, as suggested by the authors Mattei & Frellsen (2018), to mitigate the effects of poor initialisation. We further investigated two different warm-up methods for MWG: an approximate MAP initialisation using stochastic gradient ascent on the log-likelihood, and LAIR. Both schemes improved over the base MWG but we found that the initialisation using LAIR generally performed better (see fig. 12 in the appendix). MWG with LAIR initialisation is denoted in fig. 4 as MWG′ (orange). We observe that with better initialisation the performance of MWG can be significantly improved, hence confirming the sensitivity of MWG to poor initialisation as discussed in section 3. However, with few exceptions MWG′ (orange) still generally performs worse than the proposed methods (pink and yellow),

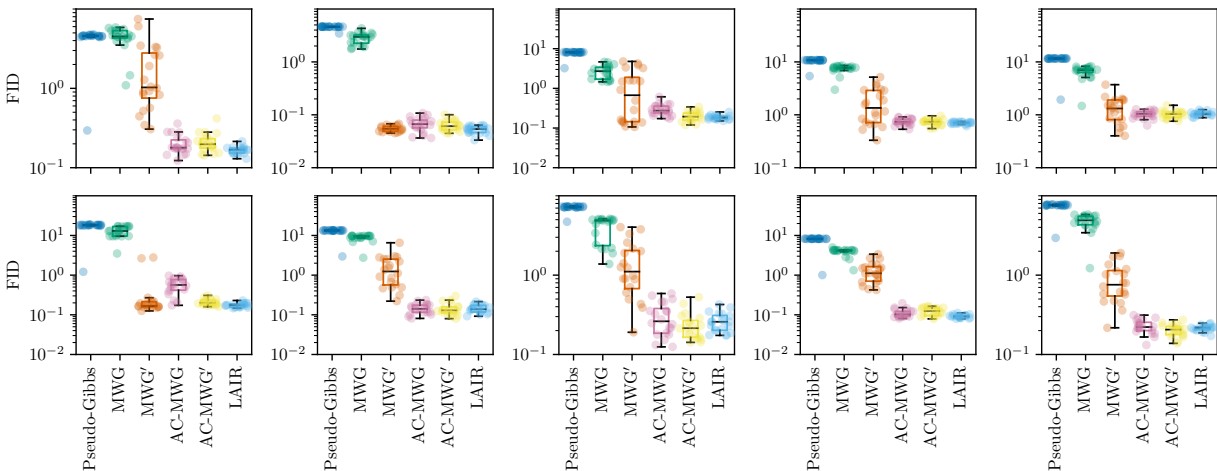

Figure 4: Fréchet inception distance (FID) between samples from the ground truth conditional $p(\boldsymbol{z} \mid \boldsymbol{x}_{\text{obs}})$, and samples from the imputation methods. Each panel in the figure corresponds to a different conditional sampling problem $p(\boldsymbol{x}_{\text{mis}} \mid \boldsymbol{x}_{\text{obs}})$. Each evaluation is repeated 20 times, and the box-plot represents the inter-quartile range, including the median, and the whiskers show the overall range of the results.

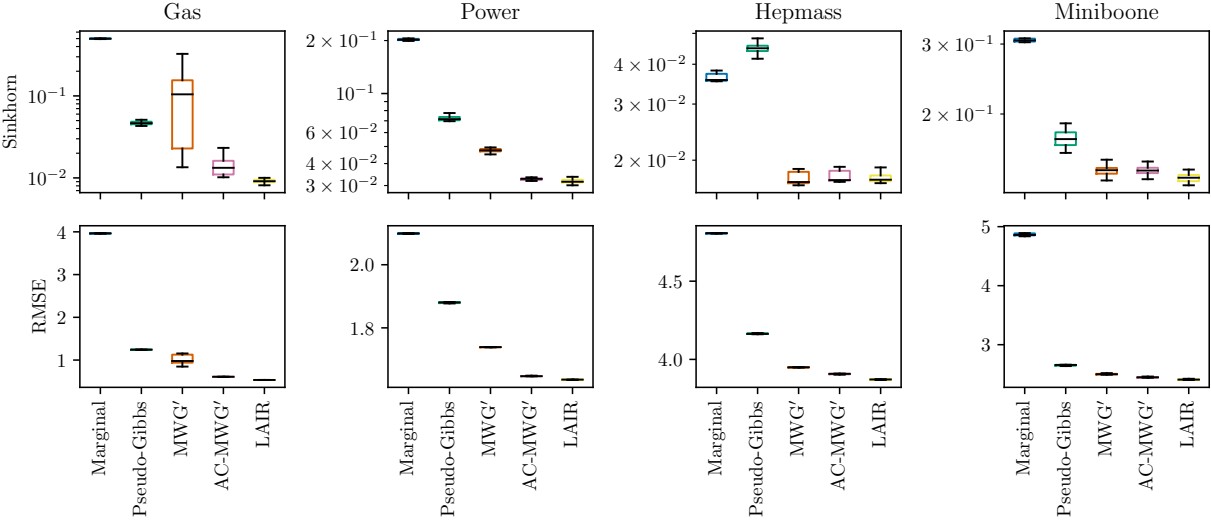

Figure 5: *Sampling performance on four real-world UCI data sets. Top:* Sinkhorn distance of the imputed data sets evaluated on a 50k data-point subset of test data (except for Miniboone where the full test data set was used). *Bottom:* Average RMSE of the imputations on the whole test data set. In both rows imputations from the final iteration of each algorithm are used and uncertainty is shown over different runs.

hence suggesting that the poor performance of MWG can be in part explained by the poor mixing of the sampler as discussed in section 3, that is addressed by the proposed methods.

## 5.2 Real-world UCI data sets

We now evaluate the proposed methods on real-world data sets from the UCI repository (Dua & Graff, 2017; Papamakarios et al., 2017). We train a VAE model with ResNet architecture on complete training data and evaluate the sampling accuracy of the existing and proposed methods on incomplete test data with 50%

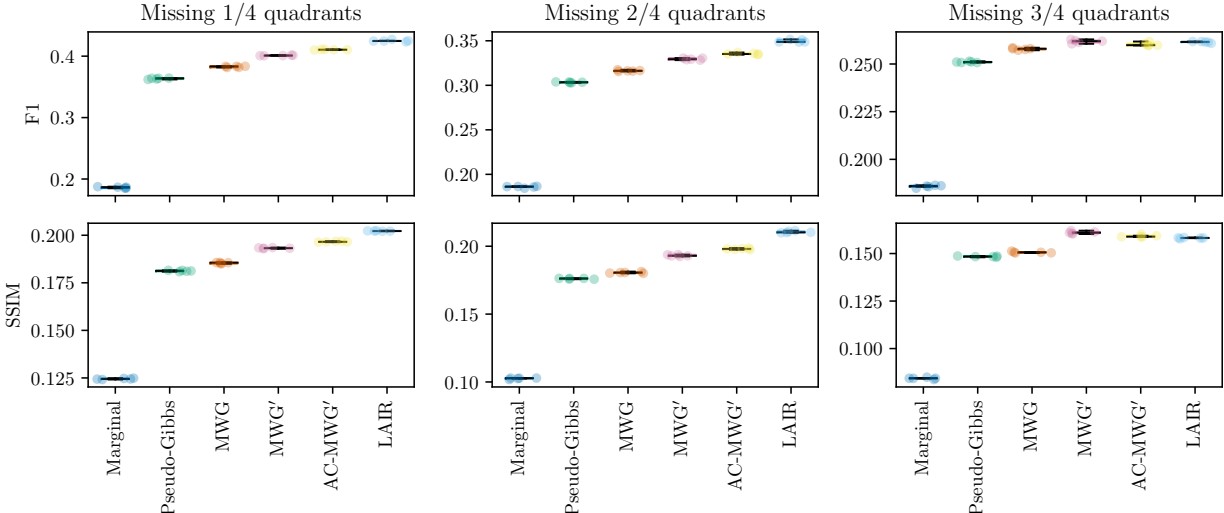

Figure 6: *Imputation accuracy on the binarised Omniglot test set with 1-3 randomly missing quadrants.* The top and bottom rows show F1 and average SSIM scores (higher is better for both metrics) respectively between the imputed and the ground truth values. In both rows imputations from the final iteration of each algorithm are used and uncertainty is shown over different runs.

missingness (see appendix C.3 for more details). We also include a simple baseline where imputations are sampled from the marginal distribution $p(\boldsymbol{x}_{\text{mis}})$ of the VAE. Moreover, in line with the observations from section 5.1 for MWG and AC-MWG we use LAIR initialisation as we have found it to considerably improve the performance of both methods. We here assess the performance using two metrics: Sinkhorn distance (Cuturi, 2013) between the imputed and ground truth data sets (computed using `geomloss` package by Feydy et al., 2019), and average RMSE of the imputations (for additional metrics, see appendix D.5).

The results are shown in fig. 5. First, the figure shows that all methods outperform marginal imputations (blue), with one exception of pseudo-Gibbs (green) on Hepmass data, where the Sinkhorn distance is slightly higher than the baseline. Second, as before, pseudo-Gibbs (green) is typically improved-upon by MWG (orange). The only exception is the Gas data with the Sinkhorn distance as metric (first row, first column) where the performance shows high variability. Other metrics (second row, first column, and appendix D.5) do not display this behaviour. Third, we see that the proposed methods, AC-MWG (pink) and LAIR (yellow), show better or comparable performance to the existing methods in terms of Sinkhorn distance (top row), and always improve on the existing methods in terms of the point-wise RMSE (bottom row). In summary, the results in this section match our findings from section 5.1, and hence further highlight the importance of mitigating the pitfalls in section 3 when dealing with real-world tasks.

## 5.3 Omniglot data set

In this section we evaluate the methods for conditional sampling of a VAE model trained on fully-observed binarised Omniglot data of handwritten characters (Lake et al., 2015). For the VAE model we use a convolutional ResNet encoder and decoder networks with 50 latent dimensions (see appendix C.4 for more details). We then evaluate the existing and proposed methods for conditional imputation of test set images that miss 1, 2, and 3 random quadrants. Similar to the previous section, we include a simple baseline where imputations are sampled from the marginal distribution $p(\boldsymbol{x}_{\text{mis}})$ of the VAE. The accuracy of the imputations on the binarised Omniglot is assessed using F1 score (Mattei & Frellsen, 2018) and structural similarity index measure (SSIM, Wang et al., 2004) between the ground truth and imputed values.

The results are shown in fig. 6. We first note that all conditional sampling methods perform better than marginal imputations (deep blue). Furthermore, we see that the metrics for the existing methods imply

the ranking pseudo-Gibbs (green) < MWG (orange) < MWG′ (pink), as before. Finally, we observe that the proposed methods, AC-MWG (yellow) and LAIR (light blue), further improve the accuracy of the imputations over the existing methods.

## 6 Discussion

Conditional sampling is a key challenge for downstream applications of VAEs and imprecise or inefficient samplers can cause unreliable results. We have examined the potential pitfalls of using Gibbs-like samplers, such as MWG, to conditionally sample from unconditional VAE models. While the outlined pitfalls are related to the well-known limitations of standard Gibbs sampler, we work out their significance in the context of VAEs. Pitfalls I and II outline two reasons for poor mixing of MWG: strong relationship between the latents $z$ and visibles $x$, and lack of exploration when the variational encoder distribution is used as proposal. Pitfall III highlights the importance of good initialisation for the performance of the sampler.

We introduced two samplers for conditional sampling of VAEs that address the pitfalls and show improved performance when compared to MWG and other baselines. The proposed methods, adaptive collapsed-Metropolis-within-Gibbs (AC-MWG) and latent-adaptive importance resampling (LAIR), mitigate pitfall I by marginalising the missing variables $x_{\text{mis}}$ when (approximately) sampling the latents $z$, and then sample the missing values $x_{\text{mis}} \sim p(x_{\text{mis}} \mid x_{\text{obs}}, z)$. Therefore, in contrast to Gibbs sampling, the two methods can be seen as approximate ancestral sampling methods with asymptotic exactness guarantees. To mitigate pitfall II we have constructed proposal distributions from a mixture composed of the variational encoder distribution and the prior, which balances exploitation and exploration. Finally, we have found that poor initialisation (pitfall III) affects LAIR much less than the MCMC methods due to its ability to use information from multiple points in the latent space, and hence using LAIR to initialise MWG and AC-MWG can further improve their respective performances.

Depending on the task, computational budget, and accuracy requirements one may choose to use either AC-MWG or LAIR for conditional sampling of VAEs. For example, in tasks where the target distribution is changing between iterations, such as learning a VAE model from incomplete data (Simkus et al., 2023), LAIR could be more efficient than AC-MWG; this is because LAIR produces valid (although potentially biased) samples from the target distribution at any iteration, while AC-MWG requires a "burn-in" period until the sampler converges to the target distribution. On the other hand, on a strict computational budget AC-MWG might be preferred over LAIR: while the cost of AC-MWG is comparable to MWG (and hence also pseudo-Gibbs), each iteration of LAIR involves equivalent computations on $K + R$ particles and hence the computational cost and memory requirements is about $K + R$ times the cost of MWG. Finally, the convergence properties of the two methods are distinct: AC-MWG converges asymptotically in number of iterations, whereas the convergence in LAIR additionally scales in the number of particles $K + R$ and therefore parallelisation may be used to improve the speed of convergence at the cost of additional memory usage.

We have focused on conditional sampling of VAE models with moderate-dimensional latent spaces. To this end, we have addressed the "exploration–exploitation" dilemma by constructing the proposal distribution from the prior and variational encoder distributions. But, what works well in moderate dimensions might not work well in high dimensions, a direct consequence of the infamous "curse of dimensionality". This means that exploring the posterior by sampling the prior distribution might become impractical in higher dimensions. To scale the methods, alternative exploration strategies could be constructed by replacing the mixture proposal in eq. (3) with, for example, a mixture composed of annealed versions of the variational encoder distribution. Moreover, since the proposed methods belong to the large and general families of adaptive MCMC (Haario et al., 2001; Warnes, 2001; Roberts & Rosenthal, 2007; Holden et al., 2009; Liang et al., 2010) and adaptive importance sampling (AIS, Cappé et al., 2004; Bugallo et al., 2017), our work opens up additional opportunities to further improve the conditional sampling of VAEs.

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

# A    AC-MWG proofs

Informally, showing convergence of MCMC samplers generally boils down to answering two questions: (i) does the Markov chain (asymptotically) reach the unique stationary distribution, and (ii) does the sampler remain in the stationary distribution after reaching it.[10]

First, we will focus on the latter question: does the AC-MWG sampler remain in the stationary distribution once it has been reached? Let $p^t$ denote the distribution after $t$ iterations, and $\pi(\boldsymbol{z}, \boldsymbol{x}_{\text{mis}}) = p(\boldsymbol{z}, \boldsymbol{x}_{\text{mis}} \mid \boldsymbol{x}_{\text{obs}})$ denote the target distribution. The following theorem formalises the answer to the question.[11]

**Theorem A.1.** *The limiting distribution of the AC-MWG sampler conditioned on the history $\mathcal{H}_{mis}^{t-1}$ is invariant, that is $p^{t-1}(\boldsymbol{z}^{t-1}, \boldsymbol{x}_{mis}^{t-1} \mid \mathcal{H}_{mis}^{t-1}) = \pi(\boldsymbol{z}^{t-1}, \boldsymbol{x}_{mis}^{t-1})$ implies $p^t(\boldsymbol{z}^t, \boldsymbol{x}_{mis}^t \mid \mathcal{H}_{mis}^t) = \pi(\boldsymbol{z}^t, \boldsymbol{x}_{mis}^t)$.*

*Proof.* Let us denote $w^t = \mathcal{H}_{\text{mis}}^t \setminus \mathcal{H}_{\text{mis}}^{t-1}$ the new variables made available in the history $\mathcal{H}_{\text{mis}}^t$ after iteration $t$ of the algorithm. Note that $w^t$ is a random variable since it depends on the accept/reject decision in lines 9 and 12 of algorithm 1. In the proof we will show that by construction of the algorithm $w^t$ and the new state $(\boldsymbol{z}^t, \boldsymbol{x}_{\text{mis}}^t)$ are independent, and that the statement in the theorem then follows.

Following algorithm 1 we now work out what are the new historical values of $w^t$ at each iteration $t$. If a proposal $\tilde{\boldsymbol{z}}$ is *rejected* then line 12 of algorithm 1 corresponds to setting $w = \varnothing$. More generally, we allow adding to the history variables that depend on the rejected state $\tilde{\boldsymbol{z}}$ but not on the current state $\boldsymbol{z}^{t-1}$. If a proposal $\tilde{\boldsymbol{z}}$ is *accepted* then line 9 of algorithm 1 corresponds to setting $w$ to be the set of imputations that were generated using the previous value of $\boldsymbol{z} = \boldsymbol{z}^{t-1}$. For instance, if new proposals were rejected for the last $r$ iterations, then $\boldsymbol{z}^{t-1-r} = \boldsymbol{z}^{t-1-r+1} = \ldots = \boldsymbol{z}^{t-1}$, and hence $\boldsymbol{x}_{\text{mis}}^{t-1-r}, \boldsymbol{x}_{\text{mis}}^{t-1-r+1}, \ldots, \boldsymbol{x}_{\text{mis}}^{t-1}$ would all depend on $\boldsymbol{z}^{t-1}$, i.e. $\boldsymbol{x}_{\text{mis}}^{t-1-r}, \boldsymbol{x}_{\text{mis}}^{t-1-r+1}, \ldots, \boldsymbol{x}_{\text{mis}}^{t-1} \sim \pi(\boldsymbol{x}_{\text{mis}} \mid \boldsymbol{z}^{t-1})$. Thus, in the case of proposal acceptance, the variable $w^t$ will contain the set of imputations $\{\boldsymbol{x}_{\text{mis}}^\tau\}_{\tau=t-1-r}^{t-1}$ that were drawn from $\pi(\boldsymbol{x}_{\text{mis}} \mid \boldsymbol{z}^{t-1})$ in the past iterations. We define the conditional distribution of $w^t$ as $\pi(w^t \mid \hat{\boldsymbol{z}}) \stackrel{!}{=} \prod_{\tau=t-1-r}^{t-1} \pi(\boldsymbol{x}_{\text{mis}}^\tau \mid \hat{\boldsymbol{z}})$ where $\hat{\boldsymbol{z}}$ is $\boldsymbol{z}^{t-1}$ if a new proposal was accepted, or $\tilde{\boldsymbol{z}}$ if a proposal was rejected. This construction of the history ensures that the proposal distribution $\tilde{q}_\epsilon(\tilde{\boldsymbol{z}} \mid \boldsymbol{x}_{\text{obs}}, \tilde{\boldsymbol{x}}_{\text{mis}})$ in lines 4 and 5 of algorithm 1 is *independent of the current $\boldsymbol{z}^{t-1}$*, and hence is a key ingredient to the proof.

We denote the transition kernel of AC-MWG as $k(\boldsymbol{z}^t, \boldsymbol{x}_{\text{mis}}^t, w^t \mid \boldsymbol{z}^{t-1}; \mathcal{H}_{\text{mis}}^{t-1})$.[12] The kernel, which depends on the history $\mathcal{H}_{\text{mis}}^{t-1}$, takes the current state of $\boldsymbol{z}^{t-1}$ and produces the new state $(\boldsymbol{z}^t, \boldsymbol{x}_{\text{mis}}^t)$ and the new historical variable $w^t$. We further use $f_{\mathcal{H}}^{t-1}(\tilde{\boldsymbol{x}}_{\text{mis}})$ to denote the probability of sampling a historical imputation $\tilde{\boldsymbol{x}}_{\text{mis}}$ from the available history $\mathcal{H}_{\text{mis}}^{t-1}$ in line 4 of algorithm 1. The kernel of AC-MWG is then defined as follows

$$
\begin{aligned}
k(\boldsymbol{z}^t, &\boldsymbol{x}_{\text{mis}}^t, w^t \mid \boldsymbol{z}^{t-1}; \mathcal{H}_{\text{mis}}^{t-1}) \\
&= \pi(\boldsymbol{x}_{\text{mis}}^t \mid \boldsymbol{z}^t) \sum_{\tilde{\boldsymbol{x}}_{\text{mis}} \in \mathcal{H}_{\text{mis}}^t} f_{\mathcal{H}}^{t-1}(\tilde{\boldsymbol{x}}_{\text{mis}}) \int \Bigg( \tilde{q}_\epsilon(\tilde{\boldsymbol{z}} \mid \boldsymbol{x}_{\text{obs}}, \tilde{\boldsymbol{x}}_{\text{mis}}) \rho_t(\tilde{\boldsymbol{z}}, \boldsymbol{z}^{t-1}; \tilde{\boldsymbol{x}}_{\text{mis}}) \delta(\boldsymbol{z}^t, \tilde{\boldsymbol{z}}) \pi(w^t \mid \boldsymbol{z}^{t-1}) \\
&\qquad\qquad\qquad\qquad + \tilde{q}_\epsilon(\tilde{\boldsymbol{z}} \mid \boldsymbol{x}_{\text{obs}}, \tilde{\boldsymbol{x}}_{\text{mis}}) \left[ 1 - \rho_t(\tilde{\boldsymbol{z}}, \boldsymbol{z}^{t-1}; \tilde{\boldsymbol{x}}_{\text{mis}}) \right] \delta(\boldsymbol{z}^t, \boldsymbol{z}^{t-1}) \pi(w^t \mid \tilde{\boldsymbol{z}}) \Bigg) \mathrm{d}\tilde{\boldsymbol{z}} \\
&= \pi(\boldsymbol{x}_{\text{mis}}^t \mid \boldsymbol{z}^t) \sum_{\tilde{\boldsymbol{x}}_{\text{mis}} \in \mathcal{H}_{\text{mis}}^t} f_{\mathcal{H}}^{t-1}(\tilde{\boldsymbol{x}}_{\text{mis}}) \Bigg( \tilde{q}_\epsilon(\boldsymbol{z}^t \mid \boldsymbol{x}_{\text{obs}}, \tilde{\boldsymbol{x}}_{\text{mis}}) \rho_t(\boldsymbol{z}^t, \boldsymbol{z}^{t-1}; \tilde{\boldsymbol{x}}_{\text{mis}}) \pi(w^t \mid \boldsymbol{z}^{t-1}) \\
&\qquad\qquad\qquad + \delta(\boldsymbol{z}^t, \boldsymbol{z}^{t-1}) \int \tilde{q}_\epsilon(\tilde{\boldsymbol{z}} \mid \boldsymbol{x}_{\text{obs}}, \tilde{\boldsymbol{x}}_{\text{mis}}) \left[ 1 - \rho_t(\tilde{\boldsymbol{z}}, \boldsymbol{z}^{t-1}; \tilde{\boldsymbol{x}}_{\text{mis}}) \right] \pi(w^t \mid \tilde{\boldsymbol{z}}) \mathrm{d}\tilde{\boldsymbol{z}} \Bigg)
\end{aligned}
$$

---

[10]The proofs in this section will consider a single observed data-point $\boldsymbol{x}_{\text{obs}}$, and hence nearly all quantities would depend on it. To ease the notation we will therefore suppress the conditioning on $\boldsymbol{x}_{\text{obs}}$ in all quantities, except for the proposal distribution $\tilde{q}_\epsilon$ in eq. (3) to keep it consistent with algorithm 1.

[11]The theorem is analogous to Theorem 1 by Holden et al. (2009) but we extend their proof to the component-wise setting of AC-MWG that involves an additional sampling step $\boldsymbol{x}_{\text{mis}}^t \sim p(\boldsymbol{x}_{\text{mis}} \mid \boldsymbol{x}_{\text{obs}}, \boldsymbol{z}^t)$, and where the history is maintained on $\boldsymbol{x}_{\text{mis}}$.

[12]Note that the kernel does not depend on the current $\boldsymbol{x}_{\text{mis}}^{t-1}$, since the new state only depends on the new $\boldsymbol{z}^t$, i.e. $\boldsymbol{x}_{\text{mis}}^t \sim \pi(\boldsymbol{x}_{\text{mis}} \mid \boldsymbol{z}^t)$.

The term in the parentheses corresponds to the standard Metropolis–Hastings kernel (see e.g. Barber, 2017, Section 27.4.2) with the addition of $w^t$ to denote the new variables to be appended to the history at iteration $t$.

Assuming that at iteration $t-1$ the sampler is already at the stationary distribution $\pi(\boldsymbol{z}^{t-1}, \boldsymbol{x}_{\mathrm{mis}}^{t-1})$, we now integrate the kernel with respect to the distribution of the current state $(\boldsymbol{z}^{t-1}, \boldsymbol{x}_{\mathrm{mis}}^{t-1})$ to obtain the marginal over $\boldsymbol{z}^t$, $\boldsymbol{x}_{\mathrm{mis}}^t$, and $w^t$

$$p^t(\boldsymbol{z}^t, \boldsymbol{x}_{\mathrm{mis}}^t, w^t \mid \mathcal{H}_{\mathrm{mis}}^{t-1}) = \int k(\boldsymbol{z}^t, \boldsymbol{x}_{\mathrm{mis}}^t, w^t \mid \boldsymbol{z}^{t-1}; \mathcal{H}_{\mathrm{mis}}^{t-1}) \pi(\boldsymbol{z}^{t-1}, \boldsymbol{x}_{\mathrm{mis}}^{t-1}) \, \mathrm{d}\boldsymbol{z}^{t-1} \, \mathrm{d}\boldsymbol{x}_{\mathrm{mis}}^{t-1}$$

Marginalising the $\boldsymbol{x}_{\mathrm{mis}}^{t-1}$

$$= \int k(\boldsymbol{z}^t, \boldsymbol{x}_{\mathrm{mis}}^t, w^t \mid \boldsymbol{z}^{t-1}; \mathcal{H}_{\mathrm{mis}}^{t-1}) \pi(\boldsymbol{z}^{t-1}) \, \mathrm{d}\boldsymbol{z}^{t-1}$$

Inserting the definition of the kernel $k$ and pushing the integral w.r.t. $\boldsymbol{z}^{t-1}$ inside the sum over $\tilde{\boldsymbol{x}}_{\mathrm{mis}}$

$$= \pi(\boldsymbol{x}_{\mathrm{mis}}^t \mid \boldsymbol{z}^t) \sum_{\tilde{\boldsymbol{x}}_{\mathrm{mis}} \in \mathcal{H}_{\mathrm{mis}}^t} f_{\mathcal{H}}^{t-1}(\tilde{\boldsymbol{x}}_{\mathrm{mis}}) \Bigg($$

$$\int \tilde{q}_\epsilon(\boldsymbol{z}^t \mid \boldsymbol{x}_{\mathrm{obs}}, \tilde{\boldsymbol{x}}_{\mathrm{mis}}) \rho_t(\boldsymbol{z}^t, \boldsymbol{z}^{t-1}; \tilde{\boldsymbol{x}}_{\mathrm{mis}}) \pi(w^t \mid \boldsymbol{z}^{t-1}) \pi(\boldsymbol{z}^{t-1}) \, \mathrm{d}\boldsymbol{z}^{t-1}$$

$$+ \int \delta(\boldsymbol{z}^t, \boldsymbol{z}^{t-1}) \int \tilde{q}_\epsilon(\tilde{\boldsymbol{z}} \mid \boldsymbol{x}_{\mathrm{obs}}, \tilde{\boldsymbol{x}}_{\mathrm{mis}}) \left[1 - \rho_t(\tilde{\boldsymbol{z}}, \boldsymbol{z}^{t-1}; \tilde{\boldsymbol{x}}_{\mathrm{mis}})\right] \pi(w^t \mid \tilde{\boldsymbol{z}}) \, \mathrm{d}\tilde{\boldsymbol{z}} \pi(\boldsymbol{z}^{t-1}) \, \mathrm{d}\boldsymbol{z}^{t-1}\Bigg)$$

Marginalising the $\boldsymbol{z}^{t-1}$ in the second integral

$$= \pi(\boldsymbol{x}_{\mathrm{mis}}^t \mid \boldsymbol{z}^t) \sum_{\tilde{\boldsymbol{x}}_{\mathrm{mis}} \in \mathcal{H}_{\mathrm{mis}}^t} f_{\mathcal{H}}^{t-1}(\tilde{\boldsymbol{x}}_{\mathrm{mis}}) \Bigg($$

$$\int \tilde{q}_\epsilon(\boldsymbol{z}^t \mid \boldsymbol{x}_{\mathrm{obs}}, \tilde{\boldsymbol{x}}_{\mathrm{mis}}) \rho_t(\boldsymbol{z}^t, \boldsymbol{z}^{t-1}; \tilde{\boldsymbol{x}}_{\mathrm{mis}}) \pi(w^t \mid \boldsymbol{z}^{t-1}) \pi(\boldsymbol{z}^{t-1}) \, \mathrm{d}\boldsymbol{z}^{t-1}$$

$$+ \int \tilde{q}_\epsilon(\tilde{\boldsymbol{z}} \mid \boldsymbol{x}_{\mathrm{obs}}, \tilde{\boldsymbol{x}}_{\mathrm{mis}}) \left[1 - \rho_t(\tilde{\boldsymbol{z}}, \boldsymbol{z}^t; \tilde{\boldsymbol{x}}_{\mathrm{mis}})\right] \pi(w^t \mid \tilde{\boldsymbol{z}}) \pi(\boldsymbol{z}^t) \, \mathrm{d}\tilde{\boldsymbol{z}}\Bigg)$$

Expanding the second summand

$$= \pi(\boldsymbol{x}_{\mathrm{mis}}^t \mid \boldsymbol{z}^t) \sum_{\tilde{\boldsymbol{x}}_{\mathrm{mis}} \in \mathcal{H}_{\mathrm{mis}}^t} f_{\mathcal{H}}^{t-1}(\tilde{\boldsymbol{x}}_{\mathrm{mis}}) \Bigg($$

$$\int \tilde{q}_\epsilon(\boldsymbol{z}^t \mid \boldsymbol{x}_{\mathrm{obs}}, \tilde{\boldsymbol{x}}_{\mathrm{mis}}) \rho_t(\boldsymbol{z}^t, \boldsymbol{z}^{t-1}; \tilde{\boldsymbol{x}}_{\mathrm{mis}}) \pi(w^t \mid \boldsymbol{z}^{t-1}) \pi(\boldsymbol{z}^{t-1}) \, \mathrm{d}\boldsymbol{z}^{t-1}$$

$$- \int \tilde{q}_\epsilon(\tilde{\boldsymbol{z}} \mid \boldsymbol{x}_{\mathrm{obs}}, \tilde{\boldsymbol{x}}_{\mathrm{mis}}) \rho_t(\tilde{\boldsymbol{z}}, \boldsymbol{z}^t; \tilde{\boldsymbol{x}}_{\mathrm{mis}}) \pi(w^t \mid \tilde{\boldsymbol{z}}) \pi(\boldsymbol{z}^t) \, \mathrm{d}\tilde{\boldsymbol{z}}$$

$$+ \pi(\boldsymbol{z}^t) \int \tilde{q}_\epsilon(\tilde{\boldsymbol{z}} \mid \boldsymbol{x}_{\mathrm{obs}}, \tilde{\boldsymbol{x}}_{\mathrm{mis}}) \pi(w^t \mid \tilde{\boldsymbol{z}}) \, \mathrm{d}\tilde{\boldsymbol{z}}\Bigg)$$

Using detailed balance $\tilde{q}_\epsilon(\tilde{\boldsymbol{z}} \mid \boldsymbol{x}_{\mathrm{obs}}, \tilde{\boldsymbol{x}}_{\mathrm{mis}}) \rho_t(\tilde{\boldsymbol{z}}, \boldsymbol{z}^t; \tilde{\boldsymbol{x}}_{\mathrm{mis}}) \pi(\boldsymbol{z}^t) = \tilde{q}_\epsilon(\boldsymbol{z}^t \mid \boldsymbol{x}_{\mathrm{obs}}, \tilde{\boldsymbol{x}}_{\mathrm{mis}}) \rho_t(\boldsymbol{z}^t, \tilde{\boldsymbol{z}}; \tilde{\boldsymbol{x}}_{\mathrm{mis}}) \pi(\tilde{\boldsymbol{z}})$ on the second summand above to obtain two identical integrals that cancel

$$= \pi(\boldsymbol{x}_{\mathrm{mis}}^t \mid \boldsymbol{z}^t) \sum_{\tilde{\boldsymbol{x}}_{\mathrm{mis}} \in \mathcal{H}_{\mathrm{mis}}^t} f_{\mathcal{H}}^{t-1}(\tilde{\boldsymbol{x}}_{\mathrm{mis}}) \Bigg($$

$$\int \tilde{q}_\epsilon(\boldsymbol{z}^t \mid \boldsymbol{x}_{\mathrm{obs}}, \tilde{\boldsymbol{x}}_{\mathrm{mis}}) \rho_t(\boldsymbol{z}^t, \boldsymbol{z}^{t-1}; \tilde{\boldsymbol{x}}_{\mathrm{mis}}) \pi(w^t \mid \boldsymbol{z}^{t-1}) \pi(\boldsymbol{z}^{t-1}) \, \mathrm{d}\boldsymbol{z}^{t-1}$$

$$- \int \tilde{q}_\epsilon(\boldsymbol{z}^t \mid \boldsymbol{x}_{\mathrm{obs}}, \tilde{\boldsymbol{x}}_{\mathrm{mis}}) \rho_t(\boldsymbol{z}^t, \tilde{\boldsymbol{z}}; \tilde{\boldsymbol{x}}_{\mathrm{mis}}) \pi(w^t \mid \tilde{\boldsymbol{z}}) \pi(\tilde{\boldsymbol{z}}) \, \mathrm{d}\tilde{\boldsymbol{z}}$$

$$+ \pi(\boldsymbol{z}^t) \int \tilde{q}_\epsilon(\tilde{\boldsymbol{z}} \mid \boldsymbol{x}_{\mathrm{obs}}, \tilde{\boldsymbol{x}}_{\mathrm{mis}}) \pi(w^t \mid \tilde{\boldsymbol{z}}) \, \mathrm{d}\tilde{\boldsymbol{z}}\Bigg)$$

Cancelling the integral terms and rearranging we obtain the marginal distribution

$$p^t(\boldsymbol{z}^t, \boldsymbol{x}_{\text{mis}}^t, w^t \mid \mathcal{H}_{\text{mis}}^{t-1}) = \pi(\boldsymbol{x}_{\text{mis}}^t, \boldsymbol{z}^t) \sum_{\tilde{\boldsymbol{x}}_{\text{mis}} \in \mathcal{H}_{\text{mis}}^t} f_{\mathcal{H}}^{t-1}(\tilde{\boldsymbol{x}}_{\text{mis}}) \int \tilde{q}_\epsilon(\tilde{\boldsymbol{z}} \mid \boldsymbol{x}_{\text{obs}}, \tilde{\boldsymbol{x}}_{\text{mis}}) \pi(w^t \mid \tilde{\boldsymbol{z}}) \, d\tilde{\boldsymbol{z}}.$$

Importantly, the factorisation shows that $(\boldsymbol{x}_{\text{mis}}^t, \boldsymbol{z}^t)$ and $w^t$ are independent, and hence

$$p^t(w^t \mid \mathcal{H}_{\text{mis}}^{t-1}) = \int p^t(\boldsymbol{z}^t, \boldsymbol{x}_{\text{mis}}^t, w^t \mid \mathcal{H}_{\text{mis}}^{t-1}) \, d\boldsymbol{z}^t \, d\boldsymbol{x}_{\text{mis}}^t = \sum_{\tilde{\boldsymbol{x}}_{\text{mis}} \in \mathcal{H}_{\text{mis}}^t} f_{\mathcal{H}}^{t-1}(\tilde{\boldsymbol{x}}_{\text{mis}}) \int \tilde{q}_\epsilon(\tilde{\boldsymbol{z}} \mid \boldsymbol{x}_{\text{obs}}, \tilde{\boldsymbol{x}}_{\text{mis}}) \pi(w^t \mid \tilde{\boldsymbol{z}}) \, d\tilde{\boldsymbol{z}}.$$

Therefore it immediately follows that

$$p^t(\boldsymbol{z}^t, \boldsymbol{x}_{\text{mis}}^t \mid \mathcal{H}_{\text{mis}}^t = \mathcal{H}_{\text{mis}}^{t-1} \cup \{w^t\}) = \frac{p^t(\boldsymbol{z}^t, \boldsymbol{x}_{\text{mis}}^t, w^t \mid \mathcal{H}_{\text{mis}}^{t-1})}{p^t(w^t \mid \mathcal{H}_{\text{mis}}^{t-1})} = \pi(\boldsymbol{x}_{\text{mis}}^t, \boldsymbol{z}^t),$$

which validates that the algorithm remains in the stationary distribution once it has reached it. □

Given that the sampler remains in the stationary distribution as shown in the above proof, we now show that the sampler can reach it. As discussed in section 4.1 the AC-MWG sampler corresponds to an ancestral sampler, which draws samples from $p(\boldsymbol{z} \mid \boldsymbol{x}_{\text{obs}})$ using non-Markovian adaptive Metropolis–Hastings, and then draws from $p(\boldsymbol{x}_{\text{mis}} \mid \boldsymbol{x}_{\text{obs}}, \boldsymbol{z})$ to obtain joint samples $(\boldsymbol{z}, \boldsymbol{x}_{\text{mis}}) \sim p(\boldsymbol{z}, \boldsymbol{x}_{\text{mis}} \mid \boldsymbol{x}_{\text{obs}})$. Therefore, to prove that the sampler reaches the stationary distribution (question (i) from the start of the section) we only need to show that the Metropolis–Hastings sampler reaches $p(\boldsymbol{z} \mid \boldsymbol{x}_{\text{obs}})$. Let $\pi(\boldsymbol{z}) = p(\boldsymbol{z} \mid \boldsymbol{x}_{\text{obs}})$ denote the target distribution, $a^t(\tilde{\boldsymbol{x}}_{\text{mis}}^t) \in [0, 1]$ a function that depends on the historical sample $\tilde{\boldsymbol{x}}_{\text{mis}}^t \sim f_{\mathcal{H}}^{t-1}(\tilde{\boldsymbol{x}}_{\text{mis}}^{t-1})$ re-sampled from the history $\mathcal{H}_{\text{mis}}^{t-1}$ in line 4 of algorithm 1 at iteration $t$, and $\tilde{\boldsymbol{X}}_{\text{mis}}^t = (\tilde{\boldsymbol{x}}_{\text{mis}}^1, \ldots, \tilde{\boldsymbol{x}}_{\text{mis}}^t)$ which denotes all those $\tilde{\boldsymbol{x}}_{\text{mis}}$ drawn up to iteration $t$, whose distribution we denote with $p_{\mathcal{H}}^t(\tilde{\boldsymbol{X}}_{\text{mis}}^t)$. We formalise the answer to question (i) in the following theorem.[13]

**Theorem A.2.** *If the likelihood of the model is bounded and the prior–variational mixture proposal in eq. (3) uses an $\epsilon > 0$, then there is a function $a^\tau(\tilde{\boldsymbol{x}}_{mis}^\tau) \in (0, 1]$ that satisfies the strong Doeblin condition*

$$\tilde{q}_\epsilon(\tilde{\boldsymbol{z}} \mid \boldsymbol{x}_{obs}, \tilde{\boldsymbol{x}}_{mis}^\tau) \geq a^\tau(\tilde{\boldsymbol{x}}_{mis}^\tau) \pi(\tilde{\boldsymbol{z}}), \quad \text{for } \forall \tilde{\boldsymbol{z}} \text{ and } \forall \tilde{\boldsymbol{x}}_{mis}^t, \tag{7}$$

*and the total variation distance is bounded from above*

$$\|p^t(\boldsymbol{z}^t) - \pi(\boldsymbol{z}^t)\|_{\text{TV}} \leq \mathbb{E}_{p_{\mathcal{H}}^t(\tilde{\boldsymbol{X}}_{mis}^t)} \left[ \prod_{\tau=1}^t (1 - a^\tau(\tilde{\boldsymbol{x}}_{mis}^\tau)) \right]. \tag{8}$$

*Hence the algorithm samples the target distribution within a finite number of iterations with a probability arbitrarily close to 1.*

*Proof.* The key observation for this proof is to note that, conditionally on the history $\mathcal{H}_{\text{mis}}^{t-1}$, each iteration $t$ of the sampler corresponds to one iteration of a (generalised) rejection sampler (see e.g. Liang et al., 2010, Section 3.1.1). Let us denote $\alpha^t \in \{0, 1\}$ a Bernoulli random variable with probability distribution $p(\alpha^t \mid \tilde{\boldsymbol{z}}, \tilde{\boldsymbol{x}}_{\text{mis}}^t) = \mathcal{B}(\alpha^t; s^t(\tilde{\boldsymbol{z}}, \tilde{\boldsymbol{x}}_{\text{mis}}^t))$ that signifies acceptance or rejection of a proposal $\tilde{\boldsymbol{z}}$ with a success probability $s^t(\tilde{\boldsymbol{z}}, \tilde{\boldsymbol{x}}_{\text{mis}}^t)$ of a rejection sampler. We obtain $s^t$ by lower-bounding the MH acceptance probability $\rho^t$ in eq. (4). We first rewrite the MH acceptance probability

$$\rho^t(\tilde{\boldsymbol{z}}, \boldsymbol{z}^{t-1}; \tilde{\boldsymbol{x}}_{\text{mis}}^t) = \min \left\{ 1, \frac{\pi(\tilde{\boldsymbol{z}})}{\pi(\boldsymbol{z}^{t-1})} \frac{\tilde{q}_\epsilon(\boldsymbol{z}^{t-1} \mid \boldsymbol{x}_{\text{obs}}, \tilde{\boldsymbol{x}}_{\text{mis}}^t)}{\tilde{q}_\epsilon(\tilde{\boldsymbol{z}} \mid \boldsymbol{x}_{\text{obs}}, \tilde{\boldsymbol{x}}_{\text{mis}}^t)} \right\}$$

$$= \frac{\pi(\tilde{\boldsymbol{z}})}{\tilde{q}_\epsilon(\tilde{\boldsymbol{z}} \mid \boldsymbol{x}_{\text{obs}}, \tilde{\boldsymbol{x}}_{\text{mis}}^t)} \min \left\{ \frac{\tilde{q}_\epsilon(\tilde{\boldsymbol{z}} \mid \boldsymbol{x}_{\text{obs}}, \tilde{\boldsymbol{x}}_{\text{mis}}^t)}{\pi(\tilde{\boldsymbol{z}})}, \frac{\tilde{q}_\epsilon(\boldsymbol{z}^{t-1} \mid \boldsymbol{x}_{\text{obs}}, \tilde{\boldsymbol{x}}_{\text{mis}}^t)}{\pi(\boldsymbol{z}^{t-1})} \right\},$$

---

[13] Our theorem here is analogous to Theorem 2 by Holden et al. (2009) but we extend the proof to the case where the proposal distribution is sampled stochastically using the history.

Lower-bounding the second term above to get $a^t(\tilde{\boldsymbol{x}}_{\text{mis}}^t)$

$$a^t(\tilde{\boldsymbol{x}}_{\text{mis}}^t) = \min_{\tilde{\boldsymbol{z}}} \min \left\{ \frac{\tilde{q}_\epsilon(\tilde{\boldsymbol{z}} \mid \boldsymbol{x}_{\text{obs}}, \tilde{\boldsymbol{x}}_{\text{mis}}^t)}{\pi(\tilde{\boldsymbol{z}})}, \frac{\tilde{q}_\epsilon(\boldsymbol{z}^{t-1} \mid \boldsymbol{x}_{\text{obs}}, \tilde{\boldsymbol{x}}_{\text{mis}}^t)}{\pi(\boldsymbol{z}^{t-1})} \right\} = \min_{\tilde{\boldsymbol{z}}} \frac{\tilde{q}_\epsilon(\tilde{\boldsymbol{z}} \mid \boldsymbol{x}_{\text{obs}}, \tilde{\boldsymbol{x}}_{\text{mis}}^t)}{\pi(\tilde{\boldsymbol{z}})}, ^{14}$$

We finally obtain a lower-bounded acceptance probability $s^t$ to the MH acceptance probability $\rho^t$

$$\rho^t(\tilde{\boldsymbol{z}}, \boldsymbol{z}^{t-1}; \tilde{\boldsymbol{x}}_{\text{mis}}^t) \geq a^t(\tilde{\boldsymbol{x}}_{\text{mis}}^t) \frac{\pi(\tilde{\boldsymbol{z}})}{\tilde{q}_\epsilon(\tilde{\boldsymbol{z}} \mid \boldsymbol{x}_{\text{obs}}, \tilde{\boldsymbol{x}}_{\text{mis}}^t)} = s^t(\tilde{\boldsymbol{z}}, \tilde{\boldsymbol{x}}_{\text{mis}}^t). ^{15}$$

We will now show that with a probability of at least $a^t(\tilde{\boldsymbol{x}}_{\text{mis}}^t)$ the sampler can jump to the stationary distribution $\pi(\boldsymbol{z})$ at any iteration $t$.

The conditional distribution of accepted samples of a rejection sampler is

$$p(\tilde{\boldsymbol{z}} \mid \tilde{\boldsymbol{x}}_{\text{mis}}^t, \alpha^t = 1) = \frac{\tilde{q}_\epsilon(\tilde{\boldsymbol{z}} \mid \boldsymbol{x}_{\text{obs}}, \tilde{\boldsymbol{x}}_{\text{mis}}^t) p(\alpha^t = 1 \mid \tilde{\boldsymbol{z}}, \tilde{\boldsymbol{x}}_{\text{mis}}^t)}{p(\alpha^t = 1 \mid \tilde{\boldsymbol{x}}_{\text{mis}}^t)} \propto \tilde{q}_\epsilon(\tilde{\boldsymbol{z}} \mid \boldsymbol{x}_{\text{obs}}, \tilde{\boldsymbol{x}}_{\text{mis}}^t) p(\alpha^t = 1 \mid \tilde{\boldsymbol{z}}, \tilde{\boldsymbol{x}}_{\text{mis}}^t).$$

Inserting $p(\alpha^t = 1 \mid \tilde{\boldsymbol{z}}, \tilde{\boldsymbol{x}}_{\text{mis}}^t) = s^t(\tilde{\boldsymbol{z}}, \tilde{\boldsymbol{x}}_{\text{mis}}^t)$ we obtain

$$p(\tilde{\boldsymbol{z}} \mid \tilde{\boldsymbol{x}}_{\text{mis}}^t, \alpha^t = 1) \propto \tilde{q}_\epsilon(\tilde{\boldsymbol{z}} \mid \boldsymbol{x}_{\text{obs}}, \tilde{\boldsymbol{x}}_{\text{mis}}^t) a^t(\tilde{\boldsymbol{x}}_{\text{mis}}^t) \frac{\pi(\tilde{\boldsymbol{z}})}{\tilde{q}_\epsilon(\tilde{\boldsymbol{z}} \mid \boldsymbol{x}_{\text{obs}}, \tilde{\boldsymbol{x}}_{\text{mis}}^t)} = a^t(\tilde{\boldsymbol{x}}_{\text{mis}}^t) \pi(\tilde{\boldsymbol{z}})$$

$$p(\alpha^t = 1 \mid \tilde{\boldsymbol{x}}_{\text{mis}}^t) = \int \tilde{q}_\epsilon(\tilde{\boldsymbol{z}} \mid \boldsymbol{x}_{\text{obs}}, \tilde{\boldsymbol{x}}_{\text{mis}}^t) p(\alpha^t = 1 \mid \tilde{\boldsymbol{z}}, \tilde{\boldsymbol{x}}_{\text{mis}}^t) \, \mathrm{d}\tilde{\boldsymbol{z}} = \int a^t(\tilde{\boldsymbol{x}}_{\text{mis}}^t) \pi(\tilde{\boldsymbol{z}}) \, \mathrm{d}\tilde{\boldsymbol{z}} = a^t(\tilde{\boldsymbol{x}}_{\text{mis}}^t)$$

Hence it follows that the accepted samples follow the target distribution

$$p(\tilde{\boldsymbol{z}} \mid \tilde{\boldsymbol{x}}_{\text{mis}}^t, \alpha^t = 1) = \frac{a^t(\tilde{\boldsymbol{x}}_{\text{mis}}^t) \pi(\tilde{\boldsymbol{z}})}{\int a^t(\tilde{\boldsymbol{x}}_{\text{mis}}^t) \pi(\hat{\boldsymbol{z}}) \, \mathrm{d}\hat{\boldsymbol{z}}} = \pi(\tilde{\boldsymbol{z}}).$$

The analogy between AC-MWG and rejection sampling allows us to conclude that the conditional probability to jump to the stationary distribution at any iteration $t$ is (at least) $a^t(\tilde{\boldsymbol{x}}_{\text{mis}}^t)$. This conditional probability depends on the historical sample $\tilde{\boldsymbol{x}}_{\text{mis}} \sim p_{\mathcal{H}}^{t-1}(\tilde{\boldsymbol{x}}_{\text{mis}})$ but is independent of the current distribution of $\boldsymbol{z}^{t-1}$.

We can now show that the probability to be in the stationary distribution within a finite number of iterations $t$ can be made arbitrarily close to 1. Let $b^t$ be the probability that the sampler *does not* jump to the stationary distribution in $t$ iterations

$$b^t(\tilde{\boldsymbol{X}}_{\text{mis}}^t) = \prod_{\tau=1}^{t} \left(1 - a^\tau(\tilde{\boldsymbol{x}}_{\text{mis}}^\tau)\right),$$

where $\tilde{\boldsymbol{X}}_{\text{mis}}^t = (\tilde{\boldsymbol{x}}_{\text{mis}}^1, \ldots, \tilde{\boldsymbol{x}}_{\text{mis}}^t)$, and let $p^t(\boldsymbol{z}^t \mid \tilde{\boldsymbol{X}}_{\text{mis}}^t)$ denote the conditional distribution of $\boldsymbol{z}^t$ after $t$ iterations

$$p^t(\boldsymbol{z}^t \mid \tilde{\boldsymbol{X}}_{\text{mis}}^t) = \pi(\boldsymbol{z}^t)(1 - b^t(\tilde{\boldsymbol{X}}_{\text{mis}}^t)) + \nu^t(\boldsymbol{z}^t \mid \tilde{\boldsymbol{X}}_{\text{mis}}^t) b^t(\tilde{\boldsymbol{X}}_{\text{mis}}^t),$$

which can be seen as a mixture of the stationary distribution $\pi(\cdot)$ with probability $(1 - b^t(\tilde{\boldsymbol{X}}_{\text{mis}}^t))$ and non-stationary distribution $\nu^t(\cdot)$ with probability $b^t(\tilde{\boldsymbol{X}}_{\text{mis}}^t)$. The marginal distribution at the $t$-th iteration is then

$$p^t(\boldsymbol{z}^t) = \int p^t(\boldsymbol{z}^t \mid \tilde{\boldsymbol{X}}_{\text{mis}}^t) p_{\mathcal{H}}^t(\tilde{\boldsymbol{X}}_{\text{mis}}^t) \, \mathrm{d}\tilde{\boldsymbol{X}}_{\text{mis}}^t$$

We now derive a bound on the total variation distance

$$\|p^t(\boldsymbol{z}^t) - \pi(\boldsymbol{z}^t)\|_{\text{TV}}$$

---

[14] Note that taking the min over $\tilde{\boldsymbol{z}}$ makes $a^t(\tilde{\boldsymbol{x}}_{\text{mis}}^t)$ independent of the current state $\boldsymbol{z}^{t-1}$!
[15] Note that due to $\rho^t \in [0,1]$ we also have that the Bernoulli success probability $s^t \in [0,1]$.

$$= \int \left| \int p^t(\boldsymbol{z}^t \mid \tilde{\boldsymbol{X}}^t_{\text{mis}}) p^t_{\mathcal{H}}(\tilde{\boldsymbol{X}}^t_{\text{mis}}) \,\mathrm{d}\tilde{\boldsymbol{X}}^t_{\text{mis}} - \pi(\boldsymbol{z}^t) \right| \mathrm{d}\boldsymbol{z}^t$$

Inserting the definition of $p^t(\boldsymbol{z}^t \mid \tilde{\boldsymbol{X}}^t_{\text{mis}})$ and using linearity of expectation to take $\pi(\boldsymbol{z}^t)$ into the expectation over $\tilde{\boldsymbol{X}}^t_{\text{mis}}$

$$= \int \left| \int \left( \pi(\boldsymbol{z}^t)(1 - b^t(\tilde{\boldsymbol{X}}^t_{\text{mis}})) + \nu^t(\boldsymbol{z}^t \mid \tilde{\boldsymbol{X}}^t_{\text{mis}}) b^t(\tilde{\boldsymbol{X}}^t_{\text{mis}}) - \pi(\boldsymbol{z}^t) \right) p^t_{\mathcal{H}}(\tilde{\boldsymbol{X}}^t_{\text{mis}}) \,\mathrm{d}\tilde{\boldsymbol{X}}^t_{\text{mis}} \right| \mathrm{d}\boldsymbol{z}^t$$

Expanding $\pi(\boldsymbol{z}^t)(1 - b^t(\tilde{\boldsymbol{X}}^t_{\text{mis}}))$ and cancelling terms

$$= \int \left| \int \left( -\pi(\boldsymbol{z}^t) + \nu^t(\boldsymbol{z}^t \mid \tilde{\boldsymbol{X}}^t_{\text{mis}}) \right) b^t(\tilde{\boldsymbol{X}}^t_{\text{mis}}) p^t_{\mathcal{H}}(\tilde{\boldsymbol{X}}^t_{\text{mis}}) \,\mathrm{d}\tilde{\boldsymbol{X}}^t_{\text{mis}} \right| \mathrm{d}\boldsymbol{z}^t$$

Applying Jensen's inequality to the (convex) norm function

$$\leq \int\!\!\int \left| -\pi(\boldsymbol{z}^t) + \nu^t(\boldsymbol{z}^t \mid \tilde{\boldsymbol{X}}^t_{\text{mis}}) \right| \mathrm{d}\boldsymbol{z}^t \, b^t(\tilde{\boldsymbol{X}}^t_{\text{mis}}) p^t_{\mathcal{H}}(\tilde{\boldsymbol{X}}^t_{\text{mis}}) \,\mathrm{d}\tilde{\boldsymbol{X}}^t_{\text{mis}}$$

Applying triangle inequality $\int |\nu(\boldsymbol{z}^t) - \pi(\boldsymbol{z}^t)| \,\mathrm{d}\boldsymbol{z}^t \leq \int |\nu(\boldsymbol{z}^t)| \,\mathrm{d}\boldsymbol{z}^t + \int |-\pi(\boldsymbol{z}^t)| \,\mathrm{d}\boldsymbol{z}^t = 2$

$$\leq 2 \int b^t(\tilde{\boldsymbol{X}}^t_{\text{mis}}) p^t_{\mathcal{H}}(\tilde{\boldsymbol{X}}^t_{\text{mis}}) \,\mathrm{d}\tilde{\boldsymbol{X}}^t_{\text{mis}} = 2\mathbb{E}_{p^t_{\mathcal{H}}(\tilde{\boldsymbol{X}}^t_{\text{mis}})} \left[ \prod_{\tau=1}^t (1 - a^\tau(\tilde{\boldsymbol{x}}^\tau_{\text{mis}})) \right]$$

Hence, the algorithm converges almost everywhere if the product goes to zero with $t \to \infty$. Therefore, if $a^\tau(\tilde{\boldsymbol{x}}^\tau_{\text{mis}})) > 0$ infinitely often then the sampler samples the target distribution $\pi(\boldsymbol{z})$ with probability arbitrarily close to 1.

To complete the proof we now show that the strong Doeblin condition (Holden, 2000; Holden et al., 2009) in eq. (7) holds, which requires that there exists $a^t(\tilde{\boldsymbol{x}}^t_{\text{mis}}) > 0$ for all $\tilde{\boldsymbol{z}}$ and $\tilde{\boldsymbol{x}}^t_{\text{mis}}$. Informally, the condition requires that the proposal distribution has heavier tails than the target distribution. We rewrite the condition in eq. (7) in its equivalent form as follows

$$\frac{\pi(\tilde{\boldsymbol{z}})}{\tilde{q}_\epsilon(\tilde{\boldsymbol{z}} \mid \boldsymbol{x}_{\text{obs}}, \tilde{\boldsymbol{x}}^t_{\text{mis}})} \leq \frac{1}{a^t(\tilde{\boldsymbol{x}}^t_{\text{mis}})}. \tag{9}$$

Inserting the definition of $\pi(\tilde{\boldsymbol{z}}) = p(\tilde{\boldsymbol{z}} \mid \boldsymbol{x}_{\text{obs}})$ and $\tilde{q}_\epsilon(\tilde{\boldsymbol{z}} \mid \boldsymbol{x}_{\text{obs}}, \tilde{\boldsymbol{x}}^t_{\text{mis}})$ from eq. (3) to the left side we obtain

$$\frac{\pi(\tilde{\boldsymbol{z}})}{\tilde{q}_\epsilon(\tilde{\boldsymbol{z}} \mid \boldsymbol{x}_{\text{obs}}, \tilde{\boldsymbol{x}}^t_{\text{mis}})} = \frac{p(\tilde{\boldsymbol{z}} \mid \boldsymbol{x}_{\text{obs}})}{(1 - \epsilon)q(\tilde{\boldsymbol{z}} \mid \boldsymbol{x}_{\text{obs}}, \tilde{\boldsymbol{x}}^t_{\text{mis}}) + \epsilon p(\tilde{\boldsymbol{z}})}$$

$$= \frac{p(\tilde{\boldsymbol{z}}, \boldsymbol{x}_{\text{obs}})}{p(\boldsymbol{x}_{\text{obs}}) \left( (1 - \epsilon)q(\tilde{\boldsymbol{z}} \mid \boldsymbol{x}_{\text{obs}}, \tilde{\boldsymbol{x}}^t_{\text{mis}}) + \epsilon p(\tilde{\boldsymbol{z}}) \right)}$$

$$= \frac{p(\boldsymbol{x}_{\text{obs}} \mid \tilde{\boldsymbol{z}})}{p(\boldsymbol{x}_{\text{obs}}) \left( (1 - \epsilon)\frac{q(\tilde{\boldsymbol{z}} \mid \boldsymbol{x}_{\text{obs}}, \tilde{\boldsymbol{x}}^t_{\text{mis}})}{p(\tilde{\boldsymbol{z}})} + \epsilon \right)}$$

$$= \frac{p(\boldsymbol{x}_{\text{obs}} \mid \tilde{\boldsymbol{z}})}{p(\boldsymbol{x}_{\text{obs}})} \left( (1 - \epsilon)\frac{q(\tilde{\boldsymbol{z}} \mid \boldsymbol{x}_{\text{obs}}, \tilde{\boldsymbol{x}}^t_{\text{mis}})}{p(\tilde{\boldsymbol{z}})} + \epsilon \right)^{-1}.$$

Hence the ratio is bounded if $\epsilon > 0$ and if the likelihood is bounded, which we can safely assume since this is already a necessary condition to well learn the model. Since the left hand side of eq. (9) is bounded it follows that $a^t(\tilde{\boldsymbol{x}}^t_{\text{mis}}) > 0$, which completes the proof.

$$\square$$

## B  Background: Importance resampling

We can generate samples following eq. (1) by using importance resampling (IR, e.g., Chopin & Papaspiliopoulos, 2020) to (approximately) sample $p(\boldsymbol{z} \mid \boldsymbol{x}_{\text{obs}})$ and then sampling $p(\boldsymbol{x}_{\text{mis}} \mid \boldsymbol{x}_{\text{obs}}, \boldsymbol{z})$ as in standard ancestral

sampling. We start with the standard importance sampling formulation for approximating the marginal $p(\boldsymbol{x}_{\text{obs}})$:

$$p(\boldsymbol{x}_{\text{obs}}) = \int p(\boldsymbol{x}_{\text{obs}}, \boldsymbol{z}) \, \mathrm{d}\boldsymbol{z} = \int q(\boldsymbol{z}) \frac{p(\boldsymbol{x}_{\text{obs}}, \boldsymbol{z})}{q(\boldsymbol{z})} \, \mathrm{d}\boldsymbol{z} = \mathbb{E}_{q(\boldsymbol{z})}\left[w(\boldsymbol{z})\right], \tag{10}$$

where $q(\boldsymbol{z})$ is a proposal distribution that is assumed easy to sample and evaluate, and $w(\boldsymbol{z}) = p(\boldsymbol{x}_{\text{obs}}, \boldsymbol{z})/q(\boldsymbol{z})$ are the (unnormalised) importance weights, which are also computationally tractable.

The importance weight function $w(\cdot)$ can then be used to re-weigh the samples from the proposal distribution $q(\boldsymbol{z})$ to follow the model posterior $p(\boldsymbol{z} \mid \boldsymbol{x}_{\text{obs}})$. We denote $\bar{w}(\tilde{\boldsymbol{z}}) = w(\tilde{\boldsymbol{z}})/\mathbb{E}_{q(\bar{\boldsymbol{z}})}\left[w(\tilde{\boldsymbol{z}})\right]$ to be the (self-)normalised importance weights, and show that samples from the proposal can be re-weighted to follow the target distribution

$$\pi(\boldsymbol{z}) = \mathbb{E}_{q(\tilde{\boldsymbol{z}})}\left[\bar{w}(\tilde{\boldsymbol{z}})\delta_{\tilde{\boldsymbol{z}}}(\boldsymbol{z})\right] = \mathbb{E}_{q(\tilde{\boldsymbol{z}})}\left[\frac{w(\tilde{\boldsymbol{z}})}{\mathbb{E}_{q(\bar{\boldsymbol{z}})}\left[w(\bar{\boldsymbol{z}})\right]}\delta_{\tilde{\boldsymbol{z}}}(\boldsymbol{z})\right] = \mathbb{E}_{q(\tilde{\boldsymbol{z}})}\left[\frac{\frac{p(\boldsymbol{x}_{\text{obs}}, \tilde{\boldsymbol{z}})}{q(\tilde{\boldsymbol{z}})}}{p(\boldsymbol{x}_{\text{obs}})}\delta_{\tilde{\boldsymbol{z}}}(\boldsymbol{z})\right] = p(\boldsymbol{z} \mid \boldsymbol{x}_{\text{obs}}), \tag{11}$$

where $\delta_{\tilde{\boldsymbol{z}}}(\cdot)$ is the Dirac delta distribution centred at point $\tilde{\boldsymbol{z}}$.

In practice, self-normalised importance resampling is generally implemented in four steps:

1. Draw M samples from a proposal $\tilde{\boldsymbol{z}}^1, \ldots, \tilde{\boldsymbol{z}}^M \sim q(\boldsymbol{z})$.

2. Compute the (unnormalised) importance weights $w(\tilde{\boldsymbol{z}}^m) = \frac{p(\boldsymbol{x}_{\text{obs}}, \tilde{\boldsymbol{z}}^m)}{q(\tilde{\boldsymbol{z}}^m)}$ for all $\forall m \in [1, M]$.

3. Self-normalise the weights $\bar{w}(\tilde{\boldsymbol{z}}^m) = \frac{w(\tilde{\boldsymbol{z}}^m)}{\sum_{l=1}^{M} w(\tilde{\boldsymbol{z}}^l)}$ for all $\forall m \in [1, M]$.

4. Resample $\boldsymbol{z}^m$ with replacement from the set $\{\tilde{\boldsymbol{z}}^m\}_{m=1}^M$ using the normalised probabilities $\bar{w}(\tilde{\boldsymbol{z}}^m)$.

Self-normalised importance sampling is consistent in the number $M$ of proposed samples and hence samples $p(\boldsymbol{z} \mid \boldsymbol{x}_{\text{obs}})$ exactly as $M \to \infty$ but has a bias of the order of $\mathcal{O}(1/M)$ (Owen, 2013; Paananen et al., 2021). Samples $\boldsymbol{x}_{\text{mis}} \sim p(\boldsymbol{x}_{\text{mis}} \mid \boldsymbol{x}_{\text{obs}})$ can then be obtained by sampling $p(\boldsymbol{x}_{\text{mis}} \mid \boldsymbol{x}_{\text{obs}}, \boldsymbol{z})$.

In standard importance sampling applications the proposal distribution $q(\boldsymbol{z})$ is traditionally chosen heuristically using the domain knowledge of the target distribution. However, in the context of VAEs specifying a good proposal can be difficult due to poor prior knowledge about the latent space of the model. Moreover, the efficiency of the sampler depends on the quality of the proposal distribution $q(\boldsymbol{z})$ and a poor proposal distribution can cause weight degeneracy in the non-asymptotic regime ($M \ll \infty$), where only a few of the proposed samples have non-zero weights, and hence poorly approximate the target distribution.

## C    Experiment details

In this appendix we provide additional details on the experiments.

### C.1    Synthetic 2D VAE

To investigate and illustrate the pitfalls of MWG we constructed a simple synthetic VAE model that approximates mixture-of-Gaussians data, see fig. 7. The visibles $\boldsymbol{x}$ are 2-dimensional and parametrised with a diagonal Gaussian decoder $p(\boldsymbol{x} \mid z)$, the latents $z$ are 1-dimensional with a uniform prior $p(z) = \text{Uniform}(0, 1)$, and the variational proposal $q(z \mid \boldsymbol{x})$ is a Beta distribution amortised with a neural network. The low-dimensional example lets us compute, via numerical integration, and visualise the conditional distributions $p(\boldsymbol{x}_{\text{mis}} \mid \boldsymbol{x}_{\text{obs}})$, $p(\boldsymbol{x}_{\text{mis}}, z \mid \boldsymbol{x}_{\text{obs}})$, and $p(z \mid \boldsymbol{x}_{\text{obs}})$. As demonstrated in the two right-most panels of fig. 7 mixing in the joint space of the missing variable and the latent $(x_0, z)$ may be poor due to low probability valleys between the modes (third panel), but could be easier in the marginal space of $z$ (last panel).

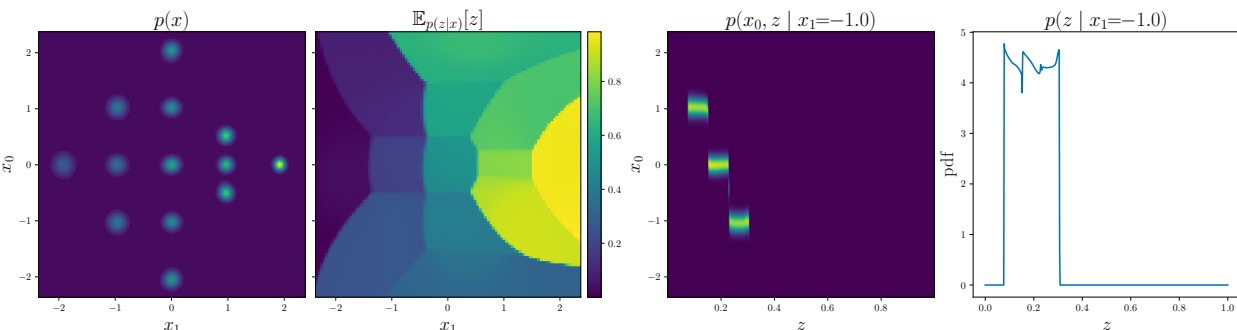

Figure 7: Left-to-right: the marginal distribution of the visibles $p(\boldsymbol{x})$ of the VAE; the posterior expected value of the latents $z$, i.e. $\mathbb{E}_{p(z|\boldsymbol{x})}[z]$; joint conditional distribution of $x_0$ and $z$ for an observed $x_1$; conditional distribution of $z$ for an observed $x_1$.

For pseudo-Gibbs and MWG in figs. 1 to 3 we perform a single run of each algorithm for 50k iterations, with both methods initialised at the same location using a sample drawn from the marginal distribution $p(\boldsymbol{x}_{\mathrm{mis}})$ of the VAE. Similarly, in fig. 2 the proposed method AC-MWG performs a single run of the algorithm for 50k iterations with mixture coefficient $\epsilon = 0.01$, initialised at the same location as pseudo-Gibbs and MWG. Finally, in fig. 3 the proposed method LAIR performs a single run of the algorithm for 2.5k iterations using 19 imputation particles ($K = 19$) and 1 replenishing mixture component ($R = 1$), the algorithm is initialised with $K = 19$ samples from the marginal distribution $p(\boldsymbol{x}_{\mathrm{mis}})$ of the VAE.

## C.2   Mixture-of-Gaussians MNIST

We construct a mixture-of-Gaussians (MoG) ground truth model with 10 multivariate Gaussian components and uniform component probability $\pi(c) = \frac{1}{10}$. Each Gaussian component is fitted on all samples from the MNIST data set (downsampled to 14x14 and transformed with a logit transformation) with a particular label $c \in [1, 10]$. We then generated a semi-synthetic data set of 18k samples and fit a VAE model with a latent space dimensionality of 25. For the VAE, we have used a diagonal Gaussian decoder using ConvResNet architecture with 4 convolutional residual blocks of feature map depths of 128, 64, 32, and 32, and a dropout of 0.2. The prior distribution over the latents is a standard normal distribution. The variational distribution is parametrised with a diagonal Gaussian encoder using ConvResNet architecture with 4 convolutional residual blocks of feature map depths 32, 64, 128, and 256, and dropout of 0.2. To optimise the VAE model we have used the sticking-the-landing gradients (Roeder et al., 2017) and fit the model using batch size of 200 for 6000 epochs using Adam optimiser (Kingma & Ba, 2014) with a learning rate of $10^{-4}$.

For pseudo-Gibbs we ran 5 independent chains for 10k iterations each, and to stabilise the sampler, the imputations were clipped to the minimum and maximum values of the data set for each dimension multiplied by 2. For MWG we have initialised 5 independent chains by running pseudo-Gibbs for 120 iterations with clipping and then run the MWG sampler for 9880 iterations on each chain. For MWG′ we have initialised 5 independent chains by running LAIR with K=4 and R=1 for 120 iterations for each chain, and then run the MWG sampler for 9880 iterations on each chain. For AC-MWG we have initialised 5 independent chains from the marginal distribution of the VAE and then run AC-MWG with $\epsilon = 0.05$ for 10k iterations. For AC-MWG′ we have initialised 5 independent chains by running LAIR with K=4 and R=1 for 120 iterations for each chain, and then run the AC-MWG sampler for 9880 iterations on each chain with $\epsilon = 0.05$. For LAIR we have initialised $K = 4$ particles from the marginal distribution of the VAE and then run the sampler with $K = 4$ and $R = 1$ for 10k iterations.

## C.3   UCI data sets

We fit VAEs on four data sets from the UCI repository (Dua & Graff, 2017) with the preprocessing of (Papamakarios et al., 2017). For all models, the variational and the generator (decoder) distributions were

fitted to be in the diagonal Gaussian family. For the encoder and decoder networks of the VAEs we fit MLP neural networks with residual block architecture using Adam optimiser (Kingma & Ba, 2014) with learning rate of $10^{-3}$ for a total of 200k stochastic gradient ascent steps (except for Miniboone where 22k steps were used) using batch size of 512 (except for Miniboone where batch size of 1024 was used), while using 8 Monte Carlo samples in each iteration to approximate the variational ELBO and sticking-the-landing gradients to reduce variance (Roeder et al., 2017). For Gas, Power, and Hepmass data the encoder and decoder networks used 2 residual blocks each with hidden dimensionality of 256, ReLU activation functions, and a latent space of 16. In addition, for Power data we add small Gaussian noise to each batch with a standard deviation of 0.001. For Miniboone data the encoder used 5 residual blocks with hidden dimensionality of 256 and decoder networks used 2 residual blocks with hidden dimensionality of 256, ReLU activation functions, a latent space of 32, and dropout of 0.5.

For pseudo-Gibbs we ran 5 independent chains for 3k iterations each, and to stabilise the sampler on Gas and Hepmass data sets imputations were clipped to the minimum and maximum values of the data set for each dimension multiplied by 2. For MWG we have initialised 5 independent chains by running LAIR with K=4 and R=1 for 100 iterations for each chain, and then run the MWG sampler for 2900 iterations on each chain. For AC-MWG we have initialised 5 independent chains by running LAIR with K=4 and R=1 for 100 iterations for each chain, and then run the AC-MWG sampler for 2900 iterations on each chain with $\epsilon = 0.3$. For LAIR we have initialised $K = 4$ particles from the marginal distribution of the VAE and then run the sampler with $K = 4$ and $R = 1$ for 3k iterations. Each method evaluations were repeated with 5 different seeds, and the uncertainty reported in the figures reflects the uncertainty over different runs.

### C.4 Handwritten character Omniglot data set

We fit a VAE on a statically binarised Omniglot data set (Lake et al., 2015) downsampled to $28 \times 28$ pixels. We have used a fixed standard Gaussian prior distribution over the latents $p(\boldsymbol{z})$ with a dimensionality of 50, an encoder distribution $q(\boldsymbol{z} \mid \boldsymbol{x})$ in the diagonal Gaussian family, and a decoder distribution $p(\boldsymbol{x} \mid \boldsymbol{z})$ in a Bernoulli family. For the encoder and decoder networks we have used convolution neural networks with ReLU activations, dropout probability of 0.2, and residual block architecture with 4 residual blocks in each networks. For the encoder the residual block hidden dimensionalities were $32, 64, 128$, and $256$, and for the decoder they were $128, 64, 32$, and $32$. We used Adam optimiser (Kingma & Ba, 2014) with a learning rate of $10^{-4}$ and a cosine annealing schedule, for a total of 3k stochastic gradient ascent steps using a batch size of 200. Moreover sticking-the-landing gradients were used to reduce encoder network gradient variance (Roeder et al., 2017).

For pseudo-Gibbs we ran 5 independent chains for 5k iterations each. For MWG we have initialised 5 independent chains by running pseudo-Gibbs for 120 iterations, and then running the MWG sampler for 4880 iterations on each chain. For MWG′ we have initialised 5 independent chains by running LAIR with K=4 and R=1 for 120 iterations for each chain, and then run the MWG sampler for 4880 iterations on each chain. For AC-MWG′ we have initialised 5 independent chains by running LAIR with K=4 and R=1 for 120 iterations for each chain, and then run AC-MWG for 4880 iterations on each chain with $\epsilon = 0.05$. For LAIR we have initialised $K = 4$ particles from the marginal distribution of the VAE and then run the sampler with $K = 4$ and $R = 1$ for 5k iterations. The above evaluations were repeated with 5 different seeds, and the uncertainty reported in the figures reflects the uncertainty over different runs.

## D  Additional figures

In this appendix we provide additional figures for the experiments in this paper.

### D.1 Synthetic 2D VAE

To aid with the understanding of the pitfalls in section 3 and our remedies in section 4, we here include additional figures on the synthetic VAE model (see details in appendix C.1). Specifically, in the top row of fig. 8 we plot the marginal distributions of the latents $p(\boldsymbol{z} \mid \boldsymbol{x}_{\text{obs}})$ that provide an additional perspective of the failure modes described in section 3: A method that is able to sample the joint distribution $p(\boldsymbol{x}_{\text{mis}}, \boldsymbol{z} \mid \boldsymbol{x}_{\text{obs}})$

must also be able to effectively sample the marginal $p(\boldsymbol{z} \mid \boldsymbol{x}_{\text{obs}})$, and if it is able to do so, then the joint $p(\boldsymbol{x}_{\text{mis}}, \boldsymbol{z} \mid \boldsymbol{x}_{\text{obs}})$ and the marginal of the missing variables $p(\boldsymbol{x}_{\text{mis}} \mid \boldsymbol{x}_{\text{obs}})$ are recovered via ancestral sampling of eq. (1).

In the left-most column (pitfall I) we can see that MWG fails to explore the unimodal posterior $p(\boldsymbol{z} \mid \boldsymbol{x}_{\text{obs}})$. As described in section 3 this is because the decoder distribution $p(\boldsymbol{x}_{\text{obs}}, \boldsymbol{x}_{\text{mis}} \mid \tilde{\boldsymbol{z}})$ places little density/mass on the *previous* value of $\boldsymbol{x}_{\text{mis}} = \boldsymbol{x}_{\text{mis}}^{t-1}$, which in turn gets such latent proposals $\tilde{\boldsymbol{z}}$ rejected. As a result, the MWG sampler remains "stuck" in a small part of the (marginal) posterior. The middle column provides an additional view of pitfall II. In particular, we see that the posterior distribution $p(\boldsymbol{z} \mid \boldsymbol{x}_{\text{obs}})$ in this case is multi-modal. However, an encoder $q(\boldsymbol{z} \mid \boldsymbol{x}_{\text{obs}}, \boldsymbol{x}_{\text{mis}})$ conditioned on a specific completed data-point $\boldsymbol{x}_{\text{obs}} \cup \boldsymbol{x}_{\text{mis}}$ is unlikely to propose a latent value $\tilde{\boldsymbol{z}}$ that would reach one of the alternative modes. As a result, the pseudo-Gibbs and MWG samplers never reach the alternative modes of the posterior $p(\boldsymbol{z} \mid \boldsymbol{x}_{\text{obs}})$, and remain stuck in a single mode. Finally, the right-most column reinforces the understanding of pitfall III. Specifically, we see that if MWG is initialised in a low-probability location of $p(\boldsymbol{z} \mid \boldsymbol{x}_{\text{obs}})$, it may fail to reach the high-probability mode.

The second and third rows of fig. 8 show the posterior approximations obtained using AC-MWG (section 4.1) and LAIR (section 4.2). As we can see, similar to the results in sections 4.1.1 and 4.2.1 the proposed methods are able to avoid the pitfalls of pseudo-Gibbs and MWG. The proposed methods remedy pitfall I by targeting the marginal $p(\boldsymbol{z} \mid \boldsymbol{x}_{\text{obs}})$ instead of the joint $p(\boldsymbol{x}_{\text{mis}}, \boldsymbol{z} \mid \boldsymbol{x}_{\text{obs}})$. Once approximate samples from the marginal $p(\boldsymbol{z} \mid \boldsymbol{x}_{\text{obs}})$ are obtained then the methods use this approximation to perform ancestral sampling of the joint $p(\boldsymbol{x}_{\text{mis}}, \boldsymbol{z} \mid \boldsymbol{x}_{\text{obs}})$. Moreover, the methods address pitfall II by using the prior–variational mixture proposals in eqs. (3) and (5), which enable exploration of the latent space. The remedy to pitfall III is related to the remedies for pitfalls I and II: the prior–variational mixture proposal enables a search of the latent space and targeting the marginal distribution $p(\boldsymbol{z} \mid \boldsymbol{x}_{\text{obs}})$ allows the sampler to move from the poor initial location to a better one by *not* conditioning on the previous imputation value $\boldsymbol{x}_{\text{mis}} = \boldsymbol{x}_{\text{mis}}^{t-1}$.

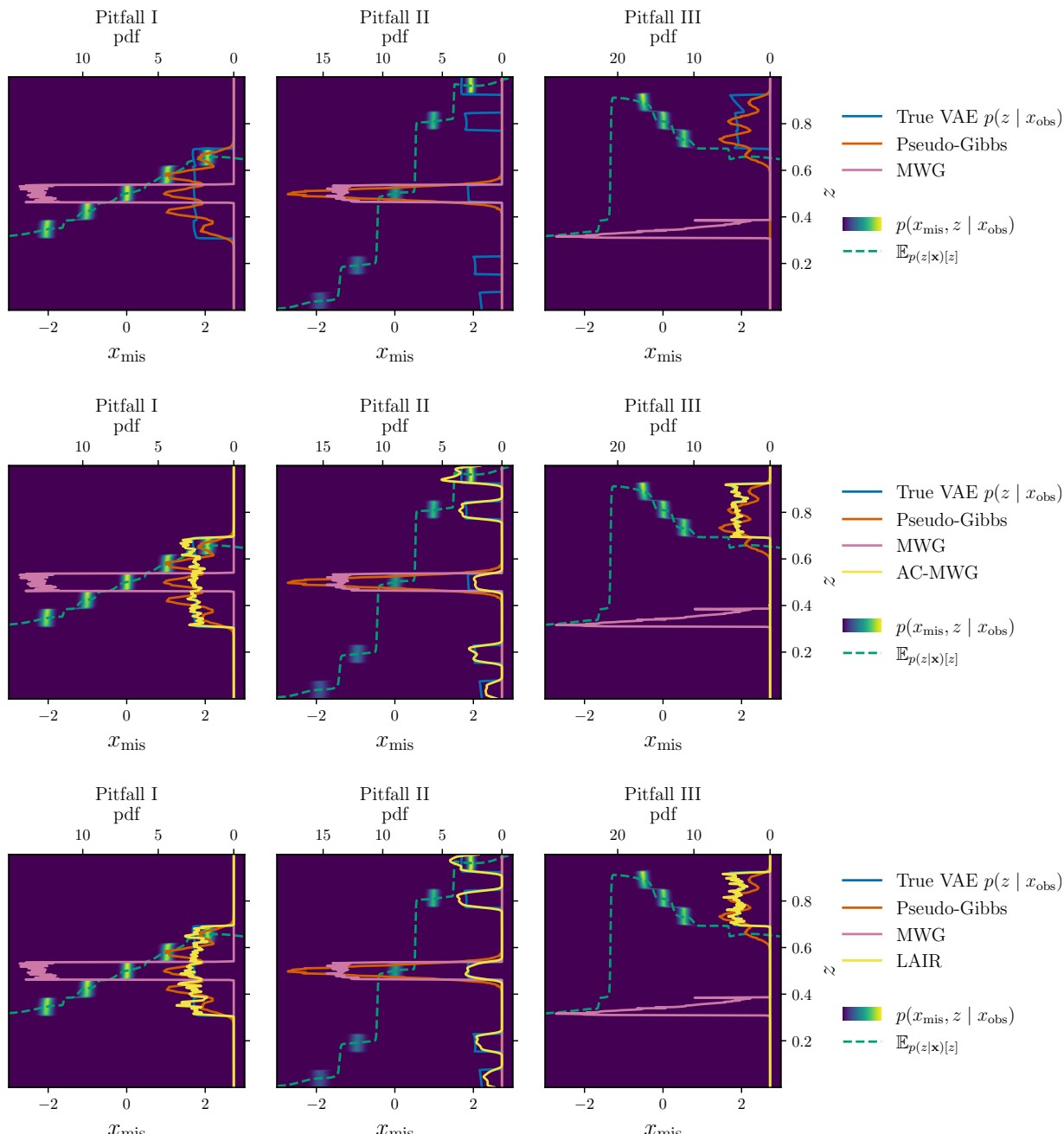

Figure 8: *Additional figures on the synthetic VAE, showing the marginals $p(z \mid x_{obs})$.* (The figure is best viewed in colour.) *Top:* showing only the true marginal $p(z \mid x_{obs})$ in blue colour and the marginals of pseudo-Gibbs (orange) and MWG (pink). *Middle:* showing the marginal of AC-MWG (yellow). *Bottom:* showing the marginal of LAIR (yellow).

## D.2 Ablation study: Synthetic 2D VAE

In this section we perform an ablation study of AC-MWG and LAIR that supplements the results in sections 4.1.1 and 4.2.1.

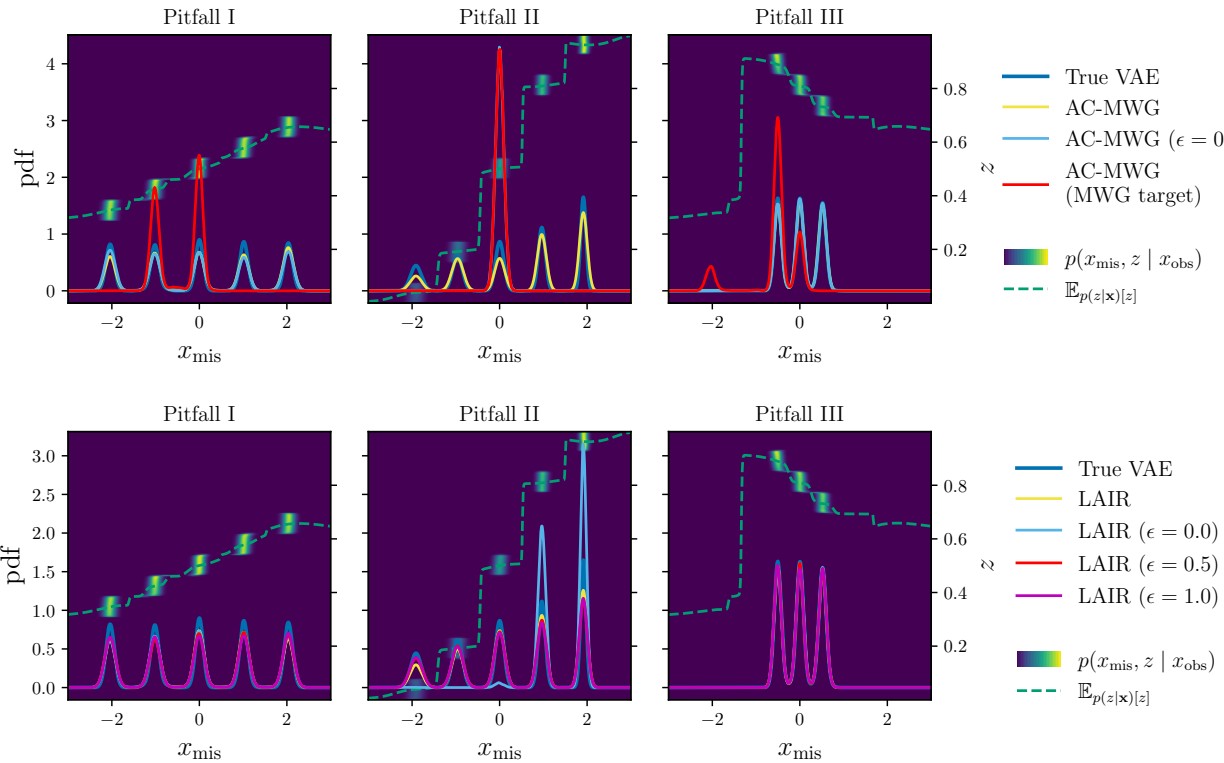

Figure 9: *Ablation studies on the 2D VAE sampling problems, same as in figs. 1 to 3.* (The figure is best viewed in colour.) *Top:* the same AC-MWG as before with $\epsilon = 0.01$ (yellow), AC-MWG with $\epsilon = 0.0$ (light blue) that does not "explore" the latent space via samples from the prior, and AC-MWG with $\epsilon = 0.01$ but the Metropolis–Hastings target of $p(z \mid x_{\text{obs}}, x_{\text{mis}})$ (red), same as MWG. *Bottom:* the same LAIR as before with $K = 19$ and $R = 1$ (i.e. $\epsilon = 0.05$, in yellow), LAIR with $K = 20$ and $R = 0$ (i.e. $\epsilon = 0.0$, in light blue) that does not "explore" the latent space using samples from the prior, LAIR with $K = 10$ and $R = 10$ (i.e. $\epsilon = 0.5$, in red), and LAIR with $K = 0$ and $R = 20$ (i.e. $\epsilon = 1.0$, in purple), which corresponds to standard (non-adaptive) importance resampling using the prior distribution as proposal.

In the top row of fig. 9 we show two ablation cases of AC-MWG. In the first case (light blue) we set $\epsilon = 0.0$ in the prior–variational mixture proposal in eq. (3). Without the prior component the sampler fails to "explore" the latent space (see the middle panel in the figure, where the light blue and red curves overlap) due to insufficiently exploratory proposal distribution (i.e. pitfall II). In the second case (red) we change the target distribution of the Metropolis–Hastings step from $p(z \mid x_{\text{obs}})$ in eq. (4) to $p(z \mid x_{\text{obs}}, x_{\text{mis}})$ used in standard MWG, that is, using the acceptance probability in eq. (2). We observe that with the MH target changed to $p(z \mid x_{\text{obs}}, x_{\text{mis}})$ the sampler also fails to mix between nearby modes in the latent space in the left-most panel (i.e. pitfall I), similar to MWG, which also affects the other two cases (middle and right panels). We therefore validate that the two modifications (the mixture proposal and the collapsed-Gibbs MH target $p(z \mid x_{\text{obs}})$) introduced in section 4.1 are key components of the AC-MWG sampler.

In the bottom row of fig. 9 we show three ablation cases of LAIR by varying the prior probability $\epsilon = \frac{R}{K+R}$ in the mixture proposal in eq. (5). The first case (light blue) corresponds to LAIR with $\epsilon = 0.0$ (or $R = 0$) and performs sub-optimally (see the middle panel) due to lack of exploration in the latent space (see pitfall II). The second case (red) corresponds to LAIR with $\epsilon = 0.5$ (or $R = K = 10$) and performs similarly to our

base LAIR case (yellow). The third case (purple) is LAIR with $\epsilon = 1.0$ and corresponds to a standard non-adaptive importance resampling with the prior distribution as the proposal. The standard importance sampling (purple) performs equally-well because of the simplicity and low-dimensionality of the latent space, however as the latent space gets more complex and higher dimensional the adaptive LAIR sampler will perform better (see results in appendix D.4).

### D.3   Mixture-of-Gaussians MNIST

Figure 10 shows the conditional mean and standard deviation at each "missing" pixel of the image, conditional on the "observed" pixels surrounded by a red border. Top-left shows the ground truth values, and the rest show values estimated from samples produced using the VAE and the (approximate) samplers. Furthermore, fig. 11 shows the absolute error in the conditional means (black is better) and signed error on the standard deviations (blue is underestimated, red is overestimated, white is perfect). The figures show a complementary view of the results in section 5.1. Interestingly, we can see that pseudo-Gibbs and MWG can overestimate the variance at some pixels while at the same time underestimating it at other pixels. The proposed methods, AC-MWG and LAIR, are less affected by this issue.

Figure 12 corresponds to fig. 4 in the main text but we additionally show MWG with MAP initialisation using stochastic gradient ascent with 5 random restarts (red). Furthermore, fig. 13 shows the experiment results using additional metrics. The additional metrics mirror the results in the main text.

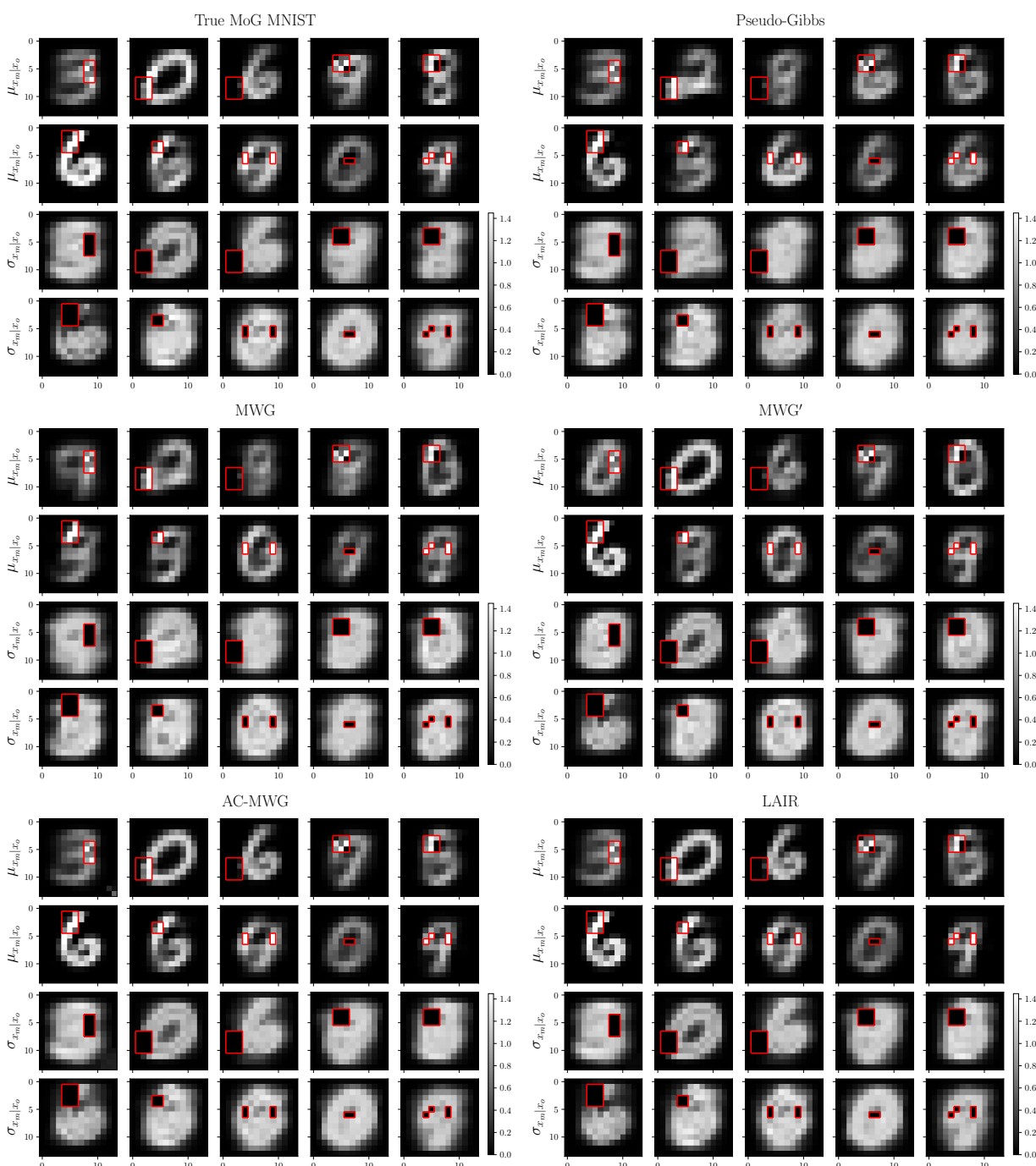

Figure 10: *Conditional mean $\mu_{\boldsymbol{x}_{mis}|\boldsymbol{x}_{obs}}$ and standard deviation $\sigma_{\boldsymbol{x}_{mis}|\boldsymbol{x}_{obs}}$ on the mixture-of-Gaussians MNIST.* The top-left panel shows the ground-truth values, and the other panels show estimates from imputations generated by the evaluated samplers. The pixels surrounded by a red border are the observed values $\boldsymbol{x}_{\text{obs}}$.

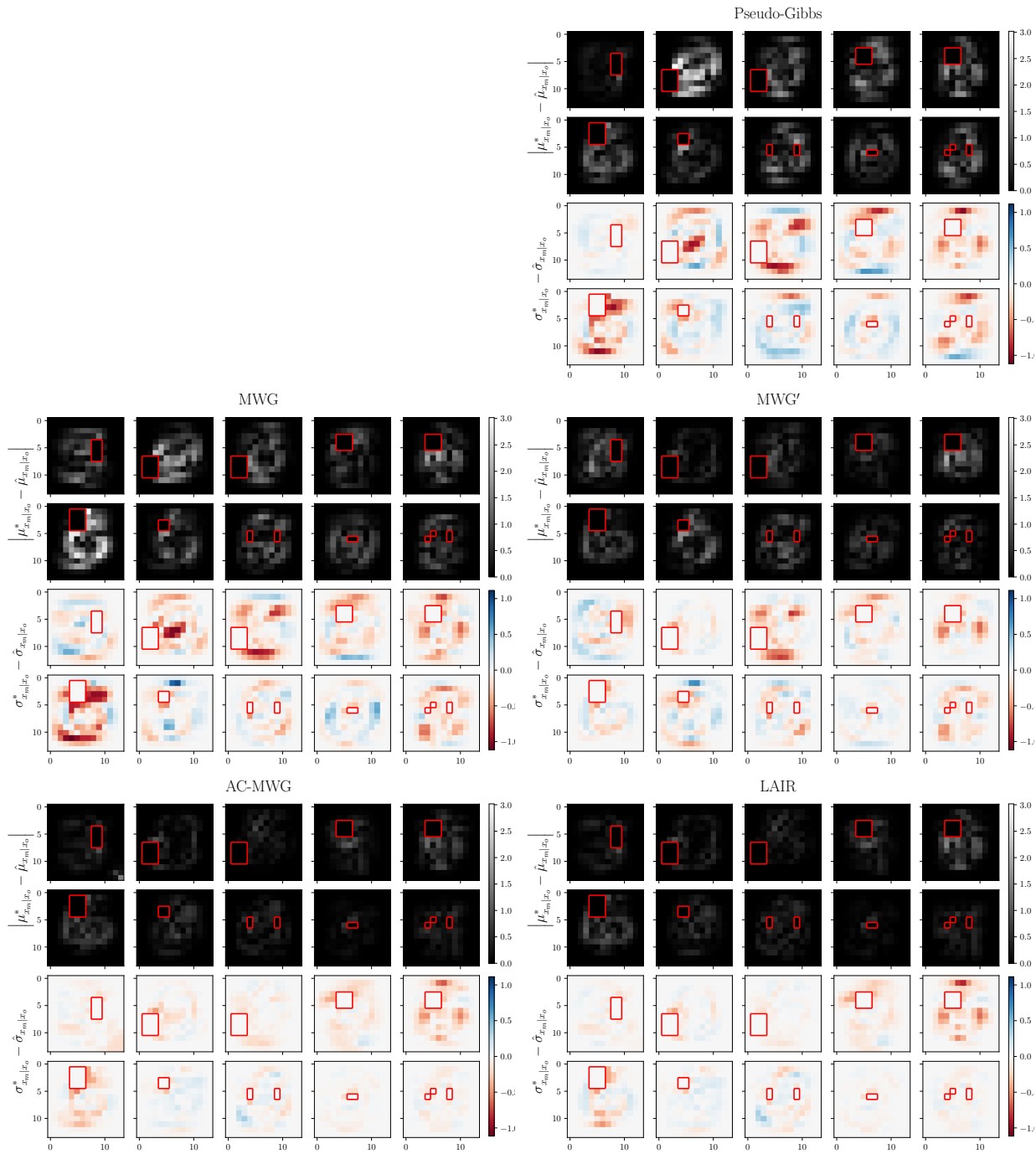

Figure 11: *The absolute error on the conditional mean $\mu_{\boldsymbol{x}_{mis}|\boldsymbol{x}_{obs}}$ and the signed error on the standard deviation $\sigma_{\boldsymbol{x}_{mis}|\boldsymbol{x}_{obs}}$ on the mixture-of-Gaussians MNIST.* We can clearly see that the proposed methods (bottom row) outperform the existing samplers.

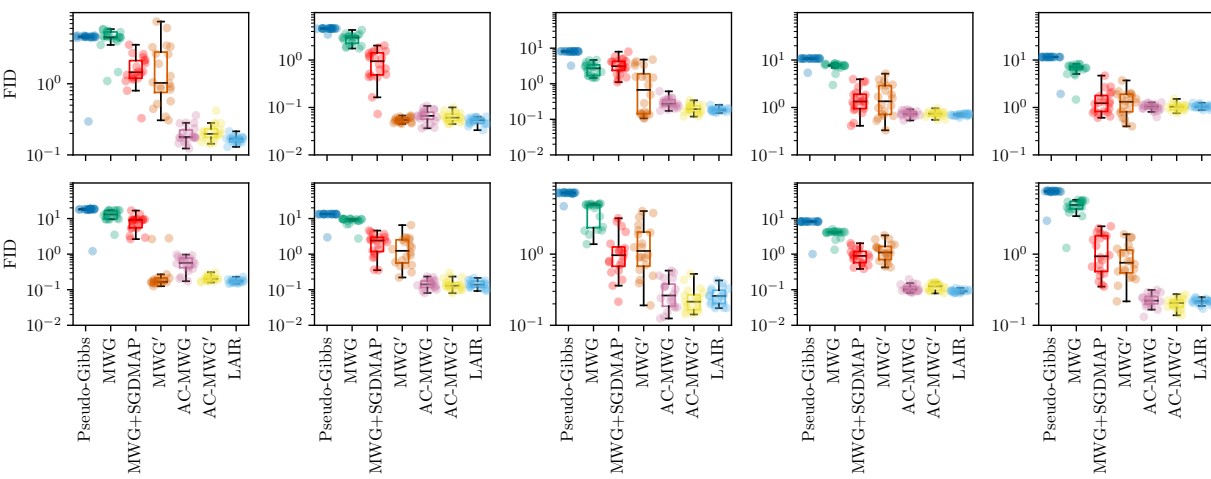

Figure 12: Same as fig. 4, but with additional method included, MWG+SGDMAP (red), which initialises MWG using stochastic gradient ascent on the log-likelihood (with 5 restarts).

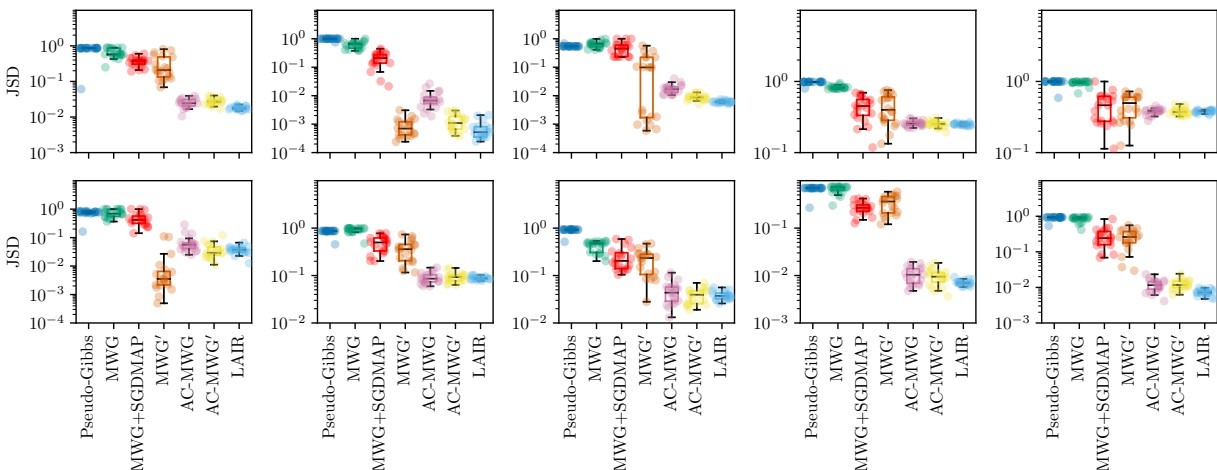

(a) Jensen–Shannon divergence (JSD) between the ground truth conditional $p(\boldsymbol{z} \mid \boldsymbol{x}_{\text{obs}})$, and estimator $\hat{p}(\boldsymbol{z} \mid \boldsymbol{x}_{\text{obs}}) = \frac{1}{N} \sum_{i=1}^{N} p(\boldsymbol{z} \mid \boldsymbol{x}_{\text{obs}}, \boldsymbol{x}_{\text{mis}}^{i})$, where $\boldsymbol{x}_{\text{mis}}^{i}$ come from the imputation methods and $N$ is the total number of imputations.

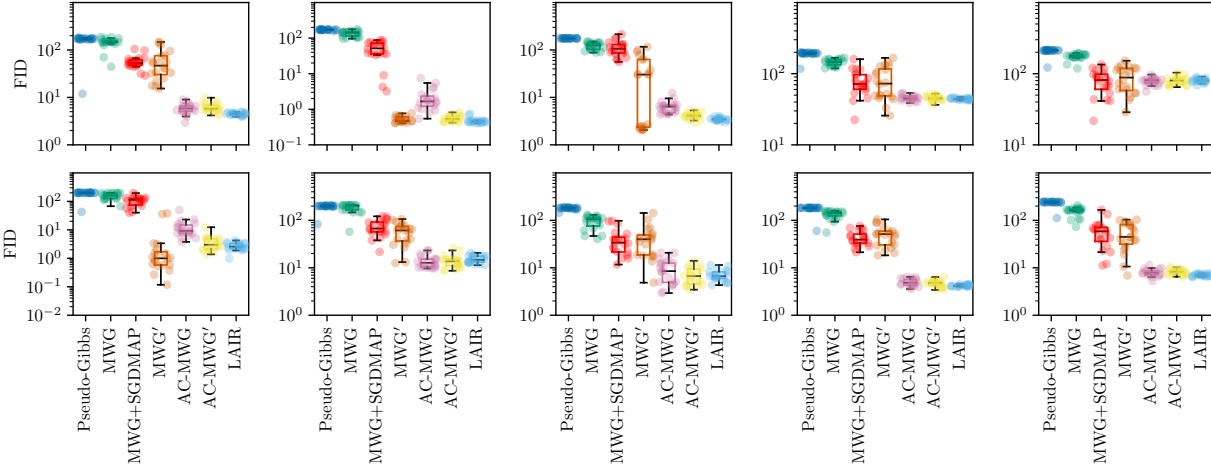

(b) Fréchet inception distance (FID) between samples from the ground truth conditional $p(\boldsymbol{z} \mid \boldsymbol{x}_{\text{obs}})$, and samples obtained from the imputation methods. The inception model features used in FID computation are the final layer outputs of a classifier neural network.

Figure 13: *Additional metrics on the mixture-of-Gaussians MNIST.* Each panel in the subfigures corresponds to a different conditional sampling problem $p(\boldsymbol{x}_{\text{mis}} \mid \boldsymbol{x}_{\text{obs}})$. Each evaluation is repeated 20 times, and the box-plot represents the inter-quartile range, including the median, and the whiskers show the overall range of the results.

## D.4 Ablation study: Mixture-of-Gaussians MNIST

This section shows an ablation study of AC-MWG and LAIR on the mixture-of-Gaussians MNIST data set that supplements the results in section 5.1.

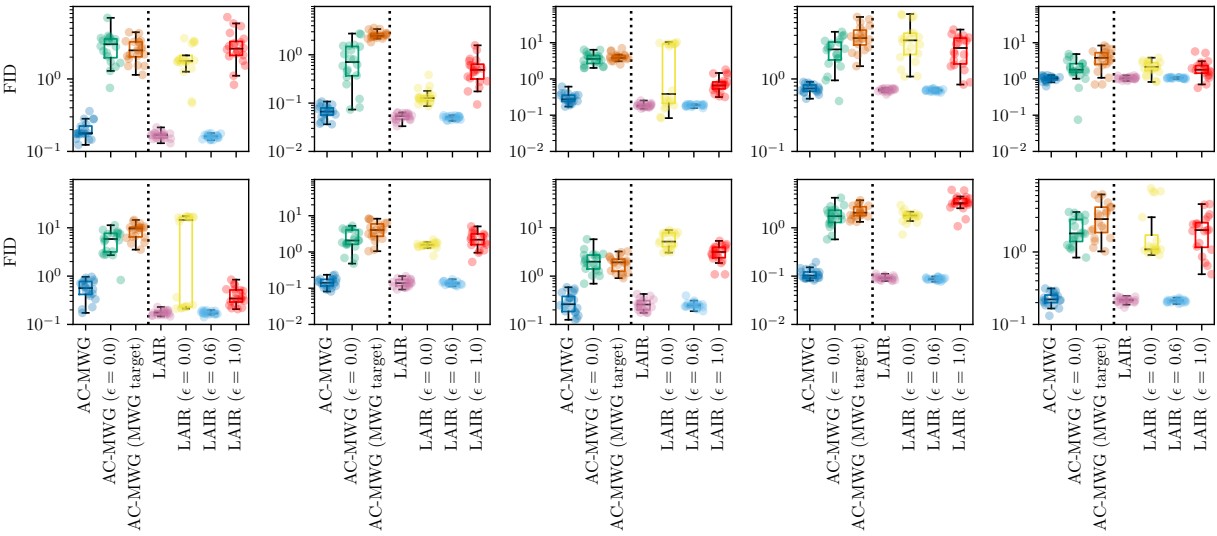

Figure 14: *Ablation studies on the MoG MNIST sampling problems, same as in fig. 4.* The FID score is computed using the final layer outputs of the encoder network as the inception features. *Left part of each panel:* AC-MWG (deep blue) is the same AC-MWG as in fig. 4 with $\epsilon = 0.05$, AC-MWG with $\epsilon = 0.0$ (green) corresponding to no prior component in the proposal distribution in eq. (3), and AC-MWG with $\epsilon = 0.05$ but the Metropolis–Hastings target of $p(\boldsymbol{z} \mid \boldsymbol{x}_{\mathrm{obs}}, \boldsymbol{x}_{\mathrm{mis}})$ (orange), same as MWG. *Right part of each panel:* LAIR (pink) is the same LAIR as in fig. 4 with $K = 4$ and $R = 1$ (i.e. $\epsilon = 0.2$), LAIR with $K = 5$ and $R = 0$ (i.e. $\epsilon = 0.0$, yellow), LAIR with $K = 2$ and $R = 3$ (i.e. $\epsilon = 0.6$, light blue), and LAIR with $K = 0$ and $R = 5$ (i.e. $\epsilon = 1.0$, red) corresponding to standard (non-adaptive) importance resampling with the prior distribution as the proposal.

The left part of each panel in fig. 14 shows two ablation cases of AC-MWG. In the first case (green) we set $\epsilon = 0.0$ in the prior–variational mixture proposal in eq. (3). As explained in pitfall II the sampler fails to explore the latent space and hence exhibits degraded performance compared to AC-MWG with $\epsilon > 0$ (deep blue). In the second case (orange) we change the target distribution of the Metropolis–Hastings step from $p(\boldsymbol{z} \mid \boldsymbol{x}_{\mathrm{obs}})$ in eq. (4) to $p(\boldsymbol{z} \mid \boldsymbol{x}_{\mathrm{obs}}, \boldsymbol{x}_{\mathrm{mis}})$ used in standard MWG, that is, using the acceptance probability in eq. (2). Similarly, we see that this ablation significantly reduces the performance of the sampler (orange versus deep blue). With this evaluation, similar to the results in appendix D.2, we validate that the proposed components (mixture proposal and collapsed-Gibbs MH target) are key to the performance of the AC-MWG method.

The right-hand part of each panel in fig. 14 shows three ablation cases of LAIR with varying prior probabilities $\epsilon = \frac{R}{K+R} \in \{0.0, 0.6, 1.0\}$ (or equivalently, varying $K$ and $R$) in the mixture proposal in eq. (5). The first case (yellow) is LAIR with $\epsilon = 0.0$ (i.e. $R = 0$), which corresponds to not using the prior distribution in the mixture proposal in eq. (5), and exhibits a significantly downgraded performance over LAIR with $\epsilon = 0.2$ (or $K = 4$ and $R = 1$, pink). The second case (light blue) is LAIR with $\epsilon = 0.6$ and performs similarly to LAIR with $\epsilon = 0.2$ (pink), hence showing that the method is not highly sensitive to the choice of $\epsilon$ as long as the edge cases ($\epsilon = 0$ and $\epsilon = 1$) are avoided. The third case (red) is LAIR with $\epsilon = 1.0$ and corresponds to a standard non-adaptive importance resampling with the prior distribution as the proposal. As we see here, the non-adaptive importance resampling (red) performs sub-optimally and hence validates that the adaptation in LAIR is important for good performance of the method.

## D.5 UCI data sets

In fig. 15 we show additional metrics of the experiments in section 5.2. We also include MWG with pseudo-Gibbs initialisation (red) as originally proposed in Mattei & Frellsen (2018). The first two rows show energy-distance MMD and Laplacian MMD between the imputed data sets and the ground truth data. We observe a similar behaviour to the results in the main text. The main exception is the Hepmass data where MWG′ (orange) seems to be preferred. However, we note that part of the good performance of MWG′ (orange) on Hepmass data is due to the use of LAIR initialisation, while using pseudo-Gibbs initialisation (red) performs similarly to LAIR (yellow). Moreover, the final row shows the average mean absolute error, and the proposed methods, AC-MWG (pink) and LAIR (yellow), are preferred over the existing methods on all data sets.

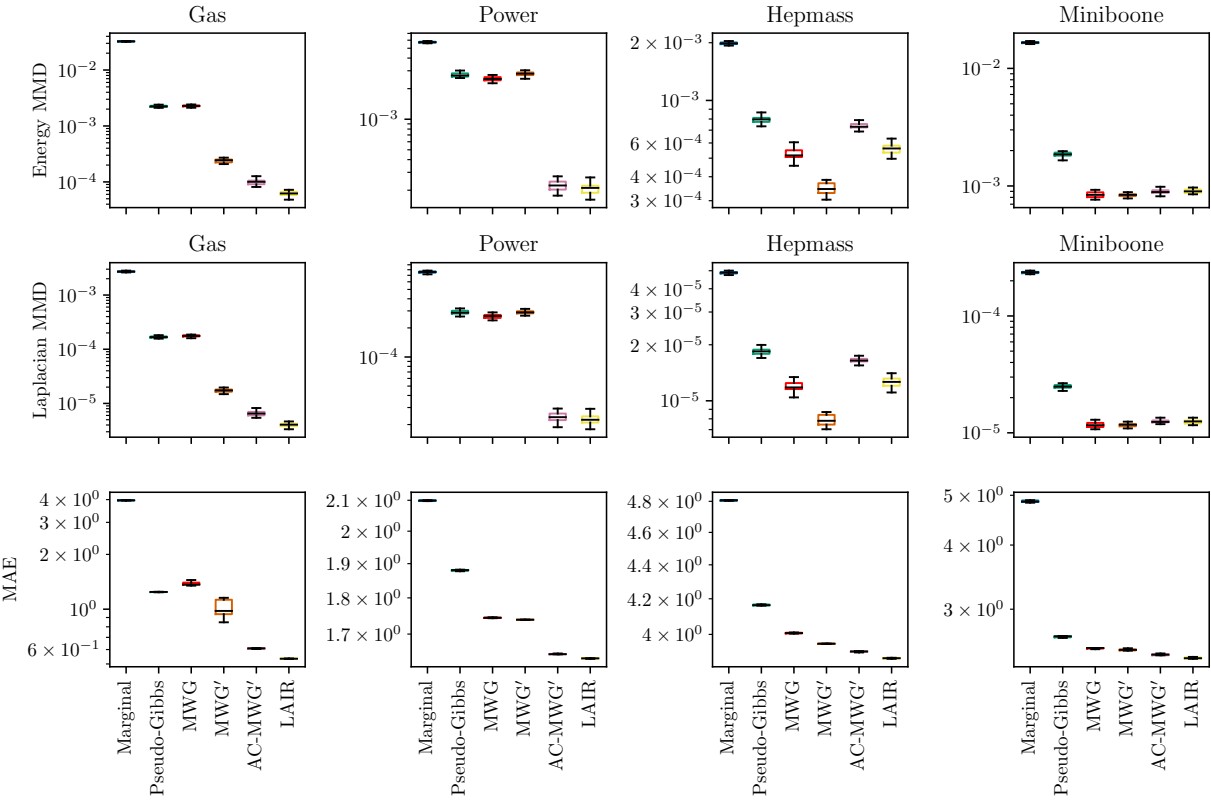

Figure 15: *Additional metrics on sampling performance on four real-world UCI data sets. Top:* energy MMD. *Middle:* Laplacian MMD. *Bottom:* average MAE of the imputations. The divergences are evaluated on a 50k data-point subset of test data (except for Miniboone where the full test data set was used), and the MAE is averaged over the full test data set. In all rows imputations from the final iteration of the corresponding algorithms are used and uncertainty is shown over different runs.

