# OpenReview forum: "Conditional Sampling of Variational Autoencoders via Iterated Approximate Ancestral Sampling"
_TMLR — Accepted by TMLR_

### Review · Reviewer_x1jP · 2023-09-15

**Summary Of Contributions:**

The paper proposes two novel methods for conditional sampling of VAEs: Adaptive collapsed Metropolis-within-Gibbs and Latent-adaptive importance resampling. These methods are motivated by three pitfalls of conditional samplers for VAEs (degenerate likelihoods causing poor mixing; encoder distributions not being wide enough; poor initialisation preventing mixing). Empirically, these methods significantly improve upon prior methods: pseudo-Gibbs and Metropolis-within-Gibbs.

**Audience:**

Yes

**Broader Impact Concerns:**

None.

**Claims And Evidence:**

Yes

**Requested Changes:**

1. (Important) Ablation studies for the methods, see the weaknesses part above.
2. (Less important, but would strengthen the paper). Exploration of how the methods behave with increasing latent space dimensionality.
3. (Minor) Ideally Figures 1&2 should be made readable in black and white print.

**Strengths And Weaknesses:**

Strengths:
1. The paper focuses on a practically important problem of drawing conditional samples in VAEs.
2. The paper is overall easy to read. I really like the structure: Section 3 introduces pitfalls of existing methods, and Section 4 fixes them with the proposed methods.
3. The proposed methods are well-motivated and work well empirically.

Weaknesses:
1. Both methods seem quite complex, while at the same time being 'modular': one could easily imagine the same methods without, say, Collapsed Gibbs. This raises the question of how practically important each piece of the methods is. Another 'module' is adaptive mixture proposal distributions. I would be very keen to see ablations for the methods.
2. All experiments in the paper have been run with latent space of 16..32 units. That's lower than 50 latent units in (Mattei & Frellsen, 2018), which makes me wonder about the scaling of the methods with respect to the latent space dimensionality. In particular, LAIR relies on importance resampling which means the bias would increase with higher dimensionality.

---

> ### Author Response · Authors · 2023-09-21
> **Response**
>
> Thank you for your comments and suggestions. We address them one by one:
>
> > Both methods seem quite complex, while at the same time being 'modular' [...] I would be very keen to see ablations for the methods.
>
> We thank the reviewer for this suggestion. In the method sections (Sections 4.1 and 4.2) we motivate each 'component' of the method based on the pitfalls identified in Section 3. Along with our exposition of the methods in Sections 4.1 and 4.2, near the text describing these components, we include back-references to the relevant pitfalls in Section 3 to help the readers understand the motivation. We agree, however, with the reviewer that an ablation study would help to better understand the _significance_ of each component. In the updated manuscript we therefore include ablation studies of the proposed methods on the synthetic VAE model (Appendix D.2) and the MoG-MNIST VAE model (Appendix D.4). We link to these appendices from Sections 4.1.1, 4.2.1, and 5.1. In these ablations we first show that both the mixture proposal and collapsed-Gibbs target for the Metropolis--Hastings step are key to the performance of AC-MWG. Second, we also investigate the sensitivity of LAIR to the choice of $\epsilon=R/(K+R)$ and observe that it performs well as long as $0 < \epsilon < 1$.
>
> > All experiments in the paper have been run with latent space of 16..32 units. That's lower than 50 latent units in (Mattei & Frellsen, 2018)...
>
> In the updated manuscript, we will include an evaluation for missing data imputation on the binarised Omniglot dataset (in the new Section 5.3). The VAE used in this experiment has 50 latent units like the one in Mattei & Frellsen (2018). The results show that the proposed methods improve the accuracy of the imputations over the existing methods.
>
> > Ideally Figures 1&2 should be made readable in black and white print.
>
> We thank the reviewer for the feedback. In the updated manuscript we will include a disclaimer in the captions of Figures 1-3 that they should be viewed in color---due to many overlapping lines we have found it difficult to produce a clear grayscale version of the figure.

---

> > ### Comment · Reviewer_x1jP · 2023-10-19
> >
> > Thank you, this addresses my concerns!
> >
> > Glad to see some evidence that LAIR method is not too sensitive to $\varepsilon$, and that both LAIR and AC-MWG scale well to more hidden units.

---

### Review · Reviewer_5zjr · 2023-10-03

**Summary Of Contributions:**

The paper proposes two sampling algorithms for the task of conditional sampling in variational autoencoders. The paper theoretically proves that the algorithms are perfectly accurate in the infinite-time limit. In finite-time, the paper numerically demonstrates that the proposed techniques alleviate pitfalls of existing ones, and gives meaningful inferences on various types of data (synthetic, UCI).

**Audience:**

Yes

**Claims And Evidence:**

Yes

**Requested Changes:**

The following are discussion questions I think the paper would benefit from (although they are not necessary):
- how to tune the number of particles K and the number of draws from the samples R in Equation 5?
  - A possible answer to this is plotting the relationship between the Frechet inception distance and K and/or R.
- are there conditions in which LAIR (Algorithm 2) is preferable to AC-MWG (Algorithm 1)?
  - in section 3, the paper nicely outlines situations in which LAIR is better than baselines (Pseudo-Gibbs or Metropolis-within-Gibbs) and AC-MWG is better than baselines.
  - what are situations in which AC-MWG is better than LAIR, in either statistical or computational sense?

**Strengths And Weaknesses:**

# Strengths
The quality of writing is very high
- The pitfalls (Section 3) are clearly explained, and the accompanying figures are helpful illustrations
- Key methodological developments (Algorithm 1, Algorithm 2) are nicely paired with motivating problems, and the paper adequately connects to previous works
- Figures (Figure 1, Figure 2 etc) are legible, and accompanying captions clearly spell out the takeaways.

The paper's numerical experiments are likely easy to replicate independently (pending release of source code after successful submission)
- Experimental setups are easy to understand
- The authors promise to release code

The paper's claims are supported by theoretical/empirical evidence
- Figure 1 and Figure 2 provide empirical evidence that the proposed algorithms can overcome the pitfalls encountered by existing methods
- Appendices contain correct proofs of the infinite-time accuracy of Algorithms 1 and Algorithm 2

# Weaknesses
None of note.

---

> ### Author Response · Authors · 2023-10-10
> **Response**
>
> Thank you for reviewing our paper and your questions.
>
> > How to tune the number of particles K and the number of draws from the samples R in Equation 5?
>
> The choice of $K$ and $R$ correspond to balancing "exploration" versus "exploitation": $\epsilon = \frac{R}{K+R}$ is the probability of sampling from the prior, making Equation 5 equivalent to a $K$-component mixture of prior--variational proposal in Equation 3. As explained in Pitfall 2, more exploration (higher $\epsilon$) may be useful when the posterior distribution $p(z \mid x_{\text{obs}})$ is highly multimodal and the encoder is incapable of proposing samples from the alternative modes, that is, the samples from the prior effectively perform a search of the alterative modes. On the other hand, if multimodality is not an issue, too much of exploration (high $\epsilon$) can result in wasteful computation due to sampling parts of the prior that have low probably under the model posterior distribution $p(z \mid x_{\text{obs}})$. Hence, the performance of LAIR and its sensitivity on the choice of $K$ and $R$ can depend on the structure of the latent space and hence the data on which the VAE was trained. With the revision we have included ablation studies for AC-MWG and LAIR on the synthetic VAE and MoG-MNIST data in Appendices D.2 and D.4. We find that, in the evaluated cases, the LAIR method performs well as long as $0 < \epsilon < 1$, meaning that we should avoid the extreme cases of using only the prior or only the variational distribution as proposal.
>
> > Are there conditions in which LAIR (Algorithm 2) is preferable to AC-MWG (Algorithm 1)?
>
> In the third paragraph of the discussion we discuss a use-case where LAIR might be preferred over AC-MWG, that is, when the target distribution $p(x_{\text{mis}} \mid x_{\text{obs}})$ changes between iterations, for example, when fitting a model using an (approximate) Monte Carlo EM.
>
> > What are situations in which AC-MWG is better than LAIR, in either statistical or computational sense?
>
> AC-MWG may be computationally preferred when memory is scarce, for example, when performing inference on large models with limited compute resources. This is because LAIR requires the evaluation of the model on $K+R$ particles, and the accuracy of the sampler may depend on it, and hence the memory cost scales with the number of particles. On the other hand, AC-MWG uses a single chain and may thus be preferred. This computational trade-off is also discussed in the third paragraph of the discussion.

---

> > ### Comment · Reviewer_5zjr · 2023-11-05
> > **Rebuttal Acknowlegment**
> >
> > I thank the reviewers for engaging with the discussion questions I brought up. I am happy with their response.

---

### Review · Reviewer_V822 · 2023-10-05

**Summary Of Contributions:**

The authors propose two methods for conditional sampling from a pre-trained VAE, where the conditional distribution implies that only a part of a test point is observed. The main drawback of previous approaches is that the sampling process can potentially be trapped in a mode and thus fail to explore the true distribution well. The proposed methods alleviate this issue by providing a better proposal distributions and a more suitable acceptance probability in an MCMC framework, and also, in an adaptive importance-sampling scheme. Theoretical results support the proposed samplers, while their efficiency is demonstrated in the experimental section.

**Audience:**

Yes

**Broader Impact Concerns:**

I think that a Broder Impact statement is not necessary as this is mainly a technical paper with no direct ethical implications.

**Claims And Evidence:**

Yes

**Requested Changes:**

Please check weaknesses. I have no additional requests for changes.

**Strengths And Weaknesses:**

**Strengths**:
- The paper is fairly well-written.
- The analysis of the previous methods is reasonable.
- The proposed methods seem to be sensible and to work well in practice.
- The theoretical analysis and the experimental results support well the claims.

**Weaknesses/Questions**:
- I think that the “pitfalls” and the “solution” can be clarified a bit more.

As regards the pitfalls:

1) If the sampled $\tilde{z}^t$ is “far” from the $\tilde{z}^{t-1}$ then the likelihood $p(x_{obs}, x_{mis}^{t-1} | \tilde{z}^t)$ is going to be low and the sample will be rejected. The problem here is that decoders imply distributions with very low entropy? (Pitfall I)

2) If the sampled $\tilde{z}^t$ is “close” to the $\tilde{z}^{t-1}$ then the likelihood $p(x_{obs}, x_{mis}^{t-1} | \tilde{z}^t)$ is high and the sample is accepted. (Pitfall II)

So both pitfalls essentially limit the exploration of the sampler in the non-assymptotic regime?

As regards the solutions:

1) The proposed proposal in both methods depends on some parameters. How sensitive are the methods with respect to the hyperparameters $\varepsilon, R, K, T$?

2) The likelihood $p(x_{obs}|\tilde{z}^t)$ is computed instead of the full joint, but if the $\tilde{z}^t$ is far from the actual code of the $x_{obs}$ the associated likelihood will be again low? Does the LAIR approach fix this by using multiple latent proposals in every step?

- The 1-dimensional examples are interesting and informative, but a bit hard to understand. Could you provide examples for 2-dimensional settings? In addition, an example to show the proposal distribution for both methods (in 2-dim) would have been very interesting. There is already Fig. 6, which I think can be improved.

- Perhaps an ablation study as regards the hyperparameters would have been beneficial to see how much they affect the performance.

---

> ### Author Response · Authors · 2023-10-10
> **Response (Part 1)**
>
> Thank you for the questions. We address them one by one:
>
> > If the sampled $\tilde z^t$ is “far” from the $\tilde z^{t-1}$ then the likelihood $p(x_{\text{obs}}, x_{\text{mis}}^{t-1} \mid {\tilde z}^t)$ is going to be low and the sample will be rejected. The problem here is that decoders imply distributions with very low entropy? (Pitfall I)
>
> This may be a misunderstanding of the Pitfall. To clarify, in Pitfall 1 we describe the failure case of MWG, where a latent proposal ${\tilde z}^t$ may be _"close"_ to the previous latent state ${z}^{t-1}$, and with the marginal posterior probability being similar for the proposed and previous states $p(\tilde z^t \mid x_{\text{obs}}) \approx p(z^{t-1} \mid x_{\text{obs}})$ (that is the $\tilde z^t$ and $z^{t-1}$ can be considered to be samples from two different posterior modes), but the proposal $\tilde z^t$ is rejected because the decoder distribution $p(x_{\text{obs}}, x_{\text{mis}^{t-1}} \mid \tilde z^t)$ places little probability on the previous imputation state $x_{\text{mis}}^{t-1}$. This pitfall motivates changing the target distribution of the Metropolis--Hastings step in MWG from $p(z \mid x_{\text{obs}}, x_{\text{mis}}^{t-1})$ to $p(z \mid x_{\text{obs}})$, similar to the collapsed-Gibbs approaches in the classical literature.
>
> > If the sampled $\tilde z^t$ is “close” to the $\tilde z^{t-1}$ then the likelihood $p(x_{\text{obs}}, x_{\text{mis}}^{t-1} \mid \tilde z^t)$ is high and the sample is accepted. (Pitfall II)
>
> This may also be a slight misunderstanding. To clarify, the Pitfall 2 is not about accepting samples where the likelihood is high but about the encoder being unable to propose latent samples $\tilde z$ that are far out from the current state $z^{t-1}$ to decode to different modes in $x$ space. As a result the sampler gets stuck in one of the modes. For example, say the encoder is trained on MNIST data and we aim to sample imputations given the upper-half of the image of a number "8". Given the upper-half of the image it may not be possible to tell if the completed image should be an "8" or a "9", so posterior samples should represent both modes. Let us assume that the current state of imputation $x_{\text{mis}}^{t-1}$ corresponds to the completion of the half-image to an image of number "9", an encoder distribution conditioned on this image will unlikely propose a $\tilde z^t$ that decodes into an image of a "8". Therefore, resulting in the sampler being stuck on producing images of "9". This pitfall therefore motivates our use of the prior--variational mixture proposal in Equations 3 and 5 _as a mean to explore the latent space_. With the revision we have included the above example in the description of Pitfall 2 in Section 3.
>
> > So both pitfalls essentially limit the exploration of the sampler in the non-assymptotic regime?
>
> Thank you for the question. Pitfalls 1 and 2 distinguish _two different failure modes_ of MWG, which limit the mixing/exploration of the sampler. The first pitfall is due to the Metropolis--Hastings target distribution used in MWG, and the second pitfall is due to the use of the encoder distribution as proposal. Importantly, however, addressing only pitfall 2 does not fix the MWG sampler, since such proposals are rejected due to pitfall 1. Conversely, addressing only pitfall 1 also does not fix the MWG sampler, since the encoder is still unable to propose larger "jumps" in the latent space. With the revision, we include ablation studies in Appendices D.2 and D.4 that show this.

---

> ### Author Response · Authors · 2023-10-10
> **Response (Part 2)**
>
> > The likelihood $p(x_{\text{obs}} \mid \tilde z^t)$ is computed instead of the full joint, but if the $\tilde z^t$ is far from the actual code of the $x_\text{obs}$ the associated likelihood will be again low? Does the LAIR approach fix this by using multiple latent proposals in every step?
>
> This is related to Pitfall 1: By using $p(x_{\text{obs}} \mid \tilde z^t)$ instead of $p(x_{\text{obs}}, x_{\text{mis}}^{t-1} \mid \tilde z^t)$ a good proposal $\tilde{z}^t$ would not be rejected due to a disagreement with the current imputation state $x_{\text{mis}}^{t-1}$. See explanation at the start of the third paragraph of Section 4.1. An alternative perspective is as follows: By using $p(x_{\text{obs}} \mid \tilde z^t)$ instead of $p(x_{\text{obs}}, x_{\text{mis}}^{t-1} \mid \tilde z^t)$ we change the target distribution from $p(z \mid x_{\text{obs}}, x_{\text{mis}}^{t-1})$ to $p(z \mid x_{\text{obs}})$. The imputations are then obtained by sampling $p(x_{\text{mis}} \mid x_{\text{obs}}, z)$. This corresponds to ancestral sampling of Equation 2. Of course, if the samplers fail to sample from $p(z \mid x_{\text{obs}})$, then the imputations will also be affected. Importantly, this issue holds for the standard MWG too because sampling of the joint $p(z, x_{\text{mis}} \mid x_{\text{obs}})$ requires that the method samples $p(z \mid x_{\text{obs}})$. This is not related to LAIR using multiple latent proposals; we need multiple samples in LAIR because any importance sampling method uses multiple weighted samples in order to represent distributions.
>
> > The 1-dimensional examples are interesting and informative, but a bit hard to understand. Could you provide examples for 2-dimensional settings? In addition, an example to show the proposal distribution for both methods (in 2-dim) would have been very interesting. There is already Fig. 6, which I think can be improved.
>
> The examples in Figures 1-3 show both the 1D distributions $p(x_{\text{mis}} \mid x_{\text{obs}})$ (coloured curves) and the 2D joint distributions $p(x_{\text{mis}}, z \mid x_{\text{obs}})$ (contour plot). These examples help us illustrate the Pitfalls 1-3 on a minimal archetypical scenario. The contour plot showing $p(x_{\text{mis}}, z \mid x_{\text{obs}})$ is important to the understanding of the pitfalls. In higher dimensions this would require a 3D plot, which would be less intelligible than the current figures. In Figure 7 (previously Figure 6) we show the details of the model used in Figures 1-3, adding a single dimension to these plots would make all of them 3D, which we believe would make the current figures more complicated.
>
> > The proposed proposal in both methods depends on some parameters. How sensitive are the methods with respect to the hyperparameters $\epsilon$, $R$, $K$, $T$?
> > Perhaps an ablation study as regards the hyperparameters would have been beneficial to see how much they affect the performance.
>
> With the revision we include two ablation studies, one on the synthetic VAE model (Appendix D.2) and one on the MoG-MNIST (Appendix D.4). In these ablations we have investigated the sensitivity of LAIR to the choice of $\epsilon=R/(K+R)$ and observed that it performs well as long as $0 < \epsilon < 1$. We also show that for AC-MWG choosing $\epsilon > 0$ as well as the use of $p(z \mid x_{\text{obs}})$ as the target distribution is important to the good performance of the method. Finally, it is important to note that the performance of the proposed methods and their sensitivity to the hyperparameters can depend on the structure of the latent space and hence the data on which the VAE was trained.

---

> > ### Comment · Reviewer_V822 · 2023-10-20
> > **Post-rebuttal**
> >
> > I would like to thank the authors for their answers and the updates to the paper. However, I think that the pitfalls and their solutions could have been described better using a graphical illustration.

---

> > > ### Author Response · Authors · 2023-10-25
> > > **Response to "Post-rebuttal"**
> > >
> > > We thank the reviewer for raising their question. We have uploaded an updated manuscript in which we provide _a new Appendix D.1_ that describes and illustrates the pitfalls from an additional point of view: illustrating the existing samplers' inability to sample the marginal $p(z \mid x_{\text{obs}})$, which consequently inhibits their ability to sample the joint $p(x_{\text{mis}}, z \mid x_{\text{obs}})$. Moreover, we discuss how the proposed methods address these issues from the alternative perspective. We believe, that this additional discussion improves the exposition of the pitfalls and helps the readers better understand the pitfalls in Section 3. We link to this appendix from Sections 3, 4.1.1 and 4.2.1.

---

> > > > ### Comment · Reviewer_V822 · 2023-10-29
> > > > **Final comments**
> > > >
> > > > I would like to thank the reviewers for taking into account the comments and updating the manuscript accordingly.

---

### Decision · Action_Editor_Kw9H · 2023-11-05

**Recommendation:** Accept as is

**Comment:**

Reviewers unanimously agreed that the paper's contribution is solid and should be published under TMLR.

The main concerns raised by the reviewers include (1) some unclear points in the discussed pitfalls, and (2) ablation studies for the components of the proposed approach. In reply, the authors added more illustrations and experiments in the updated manuscript, and the reviewers are satisfied with the revision. Therefore I suggest this paper can be accepted in its current revised form.

**Audience:**

Machine learning researchers interested in deep generative models, missing data imputation, and Bayesian computation & MCMC.

**Claims And Evidence:**

The paper focuses on conditional sampling from a pre-trained VAE which mainly has applications in missing data imputation tasks.

The contributions of the paper are:
1. Extensive discussion on previous approaches based on Gibbs sampling and the pitfalls of these methods.
2. A proposed remedy with improved Gibbs sampling for the discussed issues: (1) improved proposal distribution and MH correction based on collapsed conditional distribution, and (2) adaptation by tracking the history of accepted sample in the Gibbs steps.

The proposed approach is supported by both theoretical analyses as well as experiments on gram-scale images and UCI datasets, with previous VAE-based missing data imputation methods as baseline.